# Hepatitis Delta Virus histone mimicry drives the recruitment of chromatin remodelers for viral RNA replication

Natali Abeywickrama-Samarakoon[1,9], Jean-Claude Cortay[1,9], Camille Sureau[2], Susanne Müller [3], Dulce Alfaiate[1,4,8], Francesca Guerrieri [1,5], Apirat Chaikuad [3], Martin Schröder[3], Philippe Merle[1,6], Massimo Levrero[1,5,6]* & Paul Dény [1,7]*

Hepatitis Delta virus (HDV) is a satellite of Hepatitis B virus with a single-stranded circular RNA genome. HDV RNA genome synthesis is carried out in infected cells by cellular RNA polymerases with the assistance of the small hepatitis delta antigen (S-HDAg). Here we show that S-HDAg binds the bromodomain (BRD) adjacent to zinc finger domain 2B (BAZ2B) protein, a regulatory subunit of BAZ2B-associated remodeling factor (BRF) ISWI chromatin remodeling complexes. shRNA-mediated silencing of BAZ2B or its inactivation with the BAZ2B BRD inhibitor GSK2801 impairs HDV replication in HDV-infected human hepatocytes. S-HDAg contains a short linear interacting motif (SLiM) KacXXR, similar to the one recognized by BAZ2B BRD in histone H3. We found that the integrity of the S-HDAg SLiM sequence is required for S-HDAg interaction with BAZ2B BRD and for HDV RNA replication. Our results suggest that S-HDAg uses a histone mimicry strategy to co-activate the RNA polymerase II-dependent synthesis of HDV RNA and sustain HDV replication.

[1] INSERM, U1052 UMR CNRS 5286, Cancer Research Center of Lyon (CRCL), 151 cours Albert Thomas, 69424 Lyon, France. [2] Laboratoire de Virologie Moléculaire, INSERM U1134, Institut National de la Transfusion Sanguine, 6 rue Alexandre Cabanel, 75739 Paris, France. [3] Structural Genomics Consortium, Buchmann Institute for Molecular Life Sciences, Johann Wolfgang Goethe-University, Max-von-Laue-Strasse 15, D-60438 Frankfurt am Main, Germany. [4] Département de Pathologie et Immunologie, Université de Genève, avenue de Champel 41, 1206 Genève, Switzerland. [5] Italian Institute of Technology (IIT) - Center for Life Nanoscience (CLNS), Sapienza University, Viale Regina Elena, 291, 00161 Rome, Italy. [6] Department of Hepatology, Hôpital de la Croix Rousse, Hospices Civils de Lyon and Université Lyon l, 103 Grande Rue de la Croix-Rousse, 69004 Lyon, France. [7] Laboratoire de Microbiologie Clinique, Groupe des Hôpitaux Universitaires de Paris - Seine Saint Denis, UFR Santé Médecine, Biologie Humaine, Université Paris 13, 125 Rue de Stalingrad, 93009 Bobigny, France. [8] Present address: Department of Infectious and Tropical Diseases, Hôpital de la Croix Rousse, Hospices Civils de Lyon and Université Lyon l, 103 Grande Rue de la Croix-Rousse, 69004 Lyon, France. [9] These authors contributed equally: Natali Abeywickrama-Samarakoon, Jean-Claude Cortay. *email: massimo.levrero@inserm.fr; paul.deny@inserm.fr

Hepatitis Delta Virus (HDV) is a satellite of the hepatitis B virus (HBV)[1]. HDV genome is a single-stranded circular RNA that codes for only one protein, the HDV antigen (HDAg), devoid of any RNA polymerase activity[2]. HDV has the ability to recruit the cellular DNA-dependent RNA polymerase II (Pol II) to replicate the viral RNA. In addition to RNA synthesis, HDV genome replication involves self-cleaving activities of de novo-synthesized RNA and ligation events to form circular single-stranded molecules of both genomic and antigenomic polarities. The circular monomers are stabilized by base-pairing of ~74% of their nucleotide sequence, which confers a pseudo-double-stranded structure referred to as rod-like[3–5]. In HDV-infected cells, HDV RNA replication leads to the production of three forms of viral RNA: the genomic HDV RNA, the antigenomic RNA replication intermediate, and the messenger RNA (mRNA) for the HDAg protein produced under two isoforms: the small and large HDAg (S-HDAg and L-HDAg, respectively). Upon infection, the genomic RNA is imported into the cell nucleus for replication. Pol II first synthesizes the HDAg mRNA from the genomic HDV RNA template for S-HDAg to be produced at an early stage of infection and be available for the HDV RNA replication process to proceed[5,6]. Antigenomic RNA synthesis occurs in the nucleolus from the genomic template and is carried out by RNA Pol I or III[5]. HDV genomic RNA is then synthesized by Pol II from the antigenomic template in the nucleoplasm[5,6]. Both genomic HDV and antigenomic HDV RNAs associate with HDAg proteins to assemble the HDV ribonucleoproteins (RNPs). However, only RNPs bearing the genomic HDV RNA are assembled with the HBV envelope proteins to form HDV virions[3,7].

S-HDAg is central to HDV RNA replication and the post-translational modifications of S-HDAg, such as R13 methylation, K72 acetylation, S177 phosphorylation, and sumoylation of multiple lysine residues, are instrumental in this process[8–11]. The precise function of each of these posttranslational modifications in the HDV life cycle is not fully understood. S-HDAg phosphorylation at S177 leads to an increased affinity for the hyper-phosphorylated form of RNA Pol II and promotes the switch from initiation to elongation for antigenomic RNA synthesis in the nucleoplasm[8]. S-HDAg acetylation is thought to modulate its interaction with HDV RNA and to have an impact on HDV replication[12]. In the current model of the HDV RNP structure, four to five octamers of HDAg proteins are wrapped by a HDV RNA molecule to form a nucleosome-like structure[13–15]. According to this model, the RNP would adopt a chromatin-like organization where the viral RNA replaces the cellular DNA as template for HDV RNA synthesis by Pol II. This process is likely to require S-HDAg acetylation and the intervention of cellular chromatin remodeling factors to create a setting compatible with RNA Pol II recruitment and activity.

In this study, affinity capture coupled to mass spectrometry (MS) led to the identification of bromodomain associated to zinc finger protein 2B (BAZ2B), the regulatory subunit of the Imitation SWItch (ISWI) chromatin remodeling BRF complexes, as a major interactant of S-HDAg in differentiated human hepatocytes. We show that BAZ2B BRD, which generally binds the K14acXXR motif in histone H3-tail, recognizes the same K72acXXR motif in S-HDAg. RNA immunoprecipitation (RIP) experiments confirm the binding of BRF remodelers to the HDV RNP in HDV-infected human hepatocytes. The BAZ2B BRD-binding-defective R75A S-HDAg mutant is greatly impaired in assisting in HDV RNA replication. Altogether, our results suggest that S-HDAg mimics histone H3 acetylation to recruit BRF complexes and RNA Pol II on the HDV RNP to sustain HDV replication.

## Results

**LC-MS/MS identifies BAZ2B as a S-HDAg interactant.** We used an affinity tag purification MS approach to identify host factors that bind specifically S-HDAg. To this end, we established a lentiviral-transduced HepaRG cell line stably expressing a recombinant Strep-Tag S-HDAg protein (ST-S-HDAg) in a doxycycline-inducible manner. Under defined culture conditions, HepaRG cells give rise to well-differentiated hepatocytes (dHepaRG), which, similar to primary human hepatocytes (PHHs), are permissive to HDV infection[16]. Optimal conditions for transgene induction were defined by dose response and kinetics experiments (Supplementary Fig. 1a, b). Western blotting analysis of subcellular fractions showed that ST-S-HDAg, such as wild-type (wt) S-HDAg, accumulates in nuclear fractions (Supplementary Fig. 1c). Immunofluorescence staining confirmed the nuclear localization of ST-S-HDAg (Supplementary Fig. 1d). The ability of ST-S-HDAg to assist HDV replication was demonstrated in a *trans*-complementation assay using the replication-defective plasmid pSVL-D2M that allows for transcription of a full-length HDV RNA defective for S-HDAg synthesis and sustains replication when a functional S-HDAg is provided in *trans*[4]. As shown in Supplementary Fig. 1e, ST-S-HDAg *trans*-complemented pSVL-D2M upon doxycycline induction in the HepaRG stable cell line (lane 2) as well as upon co-transfection in HepaRG cells (lane 4). Similar results were obtained in Huh7 cells co-transfected with pSVL-D2M and pEXPR-ST-S-HDAg (Supplementary Fig. 1f, lane 2).

Using a resin-immobilized Strep-Tactin®, we performed a single-step pull-down of the recombinant ST-S-HDAg protein and its associated partners from nuclear extracts of ST-S-HDAg-expressing dHepaRG cells. The assay was carried out in the presence of Benzonase to eliminate all nucleic acid-mediated protein interactions. The processed affinity eluates were subjected to liquid chromatography coupled to tandem MS (LC-MS/MS) and the resulting peptide mass fingerprint data were searched in the Swiss-Prot human protein sequence database using the Mascot software search engine (www.matrixscience.com). The screening identified 270 proteins with a score >20 and confirmed 15 proteins previously reported to interact with S-HDAg in HEK293 cells expressing a Flag-Tag-S-HDAg bait[17] (www.imexconsortium.org; identifier IM-27520). BAZ2B appeared among the proteins with the highest Mascot score co-purifying with S-HDAg and was hence chosen for further studies. The peptide mass fingerprinting analysis identified 24 unique peptides spanning across the full length of the BAZ2B protein (Supplementary Fig. 2a). BAZ2B has been recently described as a novel ISWI regulatory subunit of the chromatin remodeler BRF due to its association with the human ATPases SNF2L/SMARCA1 (BRF-1 complex) and SNF2H/SMARCA5 (BRF-5 complex) (Supplementary Fig. 3)[18]. MS analysis also detected 6 and 11 unique tryptic peptides spanning the full-length SNF2L (P28370) and SNF2H (O60264) proteins, respectively. Four additional peptides were present in both SNF2L and SNF2H proteins (Supplementary Fig. 2b). These results suggest that, under our LC-MS/MS experimental conditions, S-HDAg can associate to both BRF-1 and BRF-5 complexes.

**S-HDAg binds BRF chromatin remodelers.** To confirm the interaction between S-HDAg and BRF complex subunits in liver cells, endogenous BAZ2B, SNF2L, and SNF2H proteins were immunoprecipitated from the nuclear extracts of Huh7 cells stably expressing wt S-HDAg. As shown in Fig. 1a, S-HDAg co-immunoprecipitated specifically with the three BRF subunits. These results show that S-HDAg is associated with both BRF-1 and BRF-5 complexes. In humans, alternative splicing generates the ATPase-dead SNF2L(ex13) variant, which contains an in-

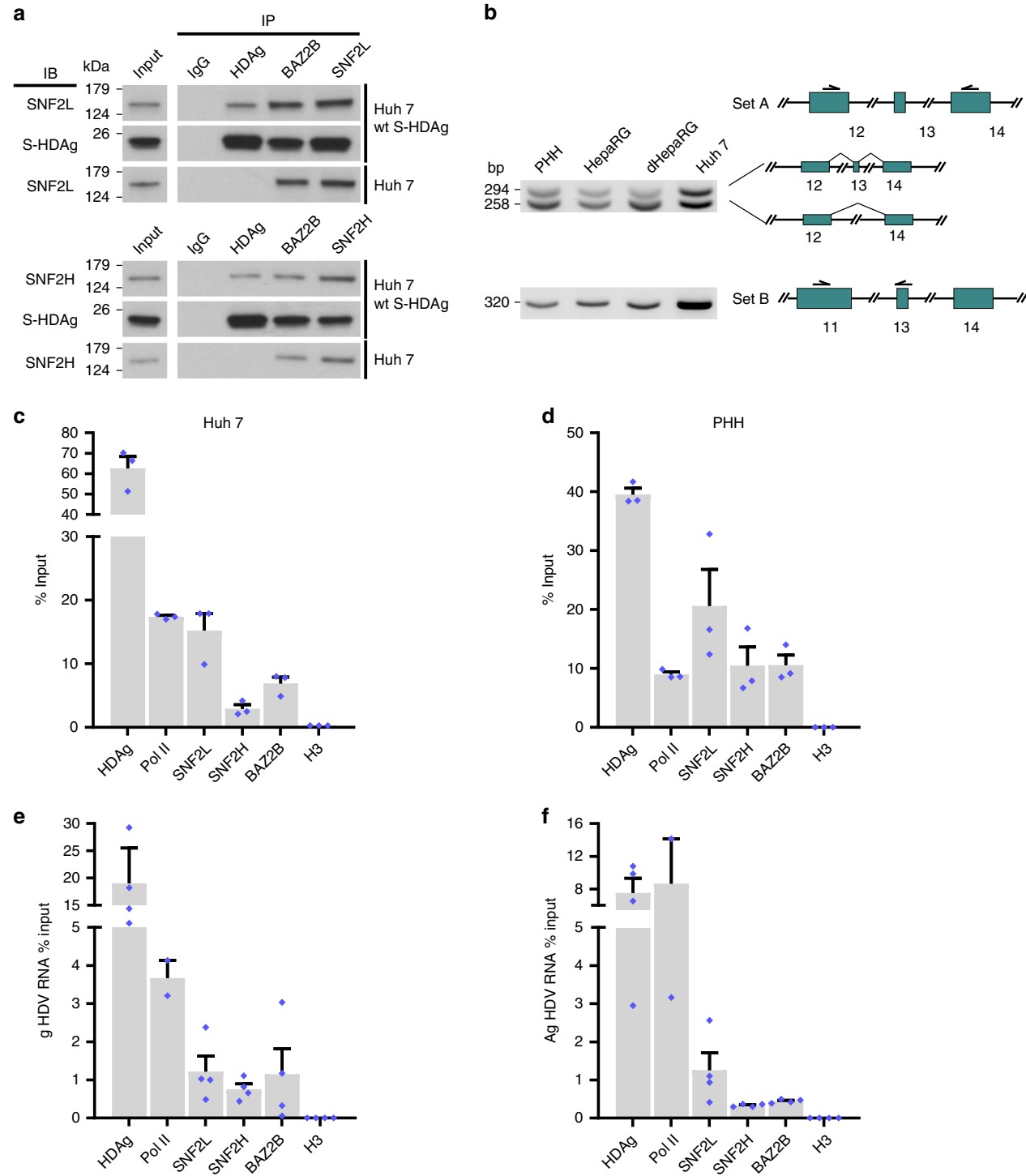

**Fig. 1 The chromatin remodeling complexes BRF-1 and BRF-5 interact with the HDV RNP. a** Interaction of wt S-HDAg with endogenous SNF2L, SNF2H, and BAZ2B. Nuclear extracts from Huh7 cells stably expressing wt S-HDAg or Huh7 cells were subjected to immunoprecipitation (IP) with α-HDAg (IP-HDAg), α-SNF2L (IP-SNF2L), α-SNF2H (IP-SNF2H), and αBAZ2B (IP-BAZ2B). Nonspecific antibody (IP-IgG) was used as a negative immunoprecipitation control. Input and elutions were analyzed by immunoblotting (IB) using the indicated antibodies. Input and HDAg lanes represent 10% and 20% volume compared with the other lanes, respectively. **b** Amplification of SNF2L variants from hepatocyte-derived cell lines. Left: PCR products demonstrating that both SNF2L and SNF2L (ex13) isoforms were observed in each cell line. Right: schematic representation of PCR products in the presence or absence of exon 13. Arrows represent primer positions. **c, d** HDV RNA immunoprecipitation assays (RIP). Glutaraldehyde-crosslinked nuclei from Huh7 cells transfected with the replication competent pSVLD3 plasmid (**c**) and PHHs infected with HDV (m.o.i. = 10) (**d**) were immunoprecipitated with antibodies directed against HDAg, RNA Pol II, SNF2L, SNF2H, and BAZ2B. HDV RNA was detected by qRT-PCR using HDV-specific primers. Histone H3 served as a negative RIP control. Error bars indicate SEMs from at least three independent experiments. **e, f** Genomic (g) or antigenomic (ag) HDV RIP assay. Glutaraldehyde-crosslinked nuclei from PHHs infected with HDV (m.o.i. = 10) were immunoprecipitated with antibodies directed against HDAg, RNA Pol II, SNF2L, SNF2H, BAZ2B, and the negative control H3. Genomic HDV (**e**) or ag HDV (**f**) RNA were detected by the biotinylated magnetic beads-based qRT-PCR assay using specific biotinylated ag HDV or g HDV primers, respectively. Values represent the mean ± SEM (n = 4 with the exception of n = 2 for Pol II RIP). Source data are provided as a Source Data file.

frame additional 36 bp exon 13 within the conserved catalytic core domain of SNF2L. This inactive variant retains the ability to incorporate itself into multiprotein complexes and is the predominant isoform in the lung, breast, and kidney, whereas the active ATPase SNF2L, bearing the nucleosome-sliding function, is expressed in brain and skeletal muscle[19]. To identify which isoform(s) of SNF2L is/are expressed in hepatocytes, we performed reverse transcriptase-PCR (RT-PCR) on total RNA extracted from PHHs, Huh7, and HepaRG cells using primer pairs (Supplementary Table 1) annealing to exons 12/14 or 11/13 in the SNF2L complementary DNA[19]. As shown in Fig. 1b, PHHs, HepaRG, and Huh7 cells predominantly express the active SNF2L together with variable levels of SNF2L(ex13). Altogether, these results indicate that, in liver cells, S-HDAg may interact with the BRF-1 complex containing the ATPase-dead SNF2L(ex13) subunit, as well as with the BRF-1 and BRF-5 complexes containing the ATPase-active SNF2L/H subunits.

**BRF-1 and BRF-5 interact with the HDV RNP**. Next, we investigated whether S-HDAg mediates the association between BRF complexes and the HDV RNP. We performed RIP experiments on glutaraldehyde-crosslinked nuclei from HDV replicating Huh7 cells transfected with the pSVLD3 plasmid and from HDV-infected PHHs. HDV RNA was highly enriched in the anti-HDAg immunoprecipitate (positive control) from HDV replicating cells, as compared with the anti-histone H3 immunoprecipitate (negative control) (Fig. 1c). HDV RNA was also pulled down specifically by antibodies directed against the BAZ2B, SNF2L, and SNF2H proteins, as well as by anti-phospho Ser5 CTD RNA Pol II antibodies (Fig. 1c). Similar results were obtained in HDV-infected PHHs (Fig. 1d). These results indicate that the BRF host chromatin remodelers associate with S-HDAg on Pol II-bound transcriptionally active/replicating HDV RNP. To assess whether we could detect differences in the recruitment of phosphorylated P-Ser5-Pol II and BRF components onto HDV RNPs containing genomic or antigenomic HDV RNA, we analyzed the RIP precipitates with biotinylated oligodeoxynucleotides designed to prime specifically the genomic HDV or the antigenomic HDV cDNA synthesis[20]. As shown in Fig. 1e/f, S-HDAg and P-Ser5 Pol II are recruited on both genomic and antigenomic HDV RNPs with similar efficiency, whereas BAZ2B and SNF2H displayed a preferential binding on the HDV genomic strand (Fig. 1e).

**BAZ2B is a host co-activator of HDV replication in PHHs**. To assess the impact of HDV RNP association with BRF complexes in HDV-infected PHHs, we employed two complementary approaches: (i) abrogating of BAZ2B expression by lentivirus-mediated transduction of specific short hairpin RNAs (shRNAs) and (ii) inhibiting BAZ2B BRD activity by the small molecule inhibitor GSK2801[21]. As shown in Fig. 2, a 40–50% reduction of BAZ2B mRNA levels (Fig. 2a) translated into a >50% inhibition of HDV replication at day 8 post infection (Fig. 2b). No cytotoxicity was observed in non-infected and HDV-infected PHHs transduced with scramble or BAZ2B shRNAs (Fig. 2c). GSK2801 treatment (10 μM) resulted in a significant reduction of HDV replication (Fig. 2d), in the absence of any significant cytotoxicity (Fig. 2e). The control compound GSK8573, which has no effect on BAZ2B BRD, did not affect HDV replication (Fig. 2d)[21]. To analyze the functional consequences of BAZ2B BRD inhibition on BRF-1 association with the HDV RNP, RIP assays were performed in HDV-infected PHHs with and without GSK2801. We observed that the RIP signal was strongly decreased with all antibodies and in particular with anti-BAZ2B and anti-SNF2L when the BAZ2B BRD activity was inhibited (Fig. 2f). These

results further confirm the involvement of BAZ2B BRD in HDV replication.

Finally, we performed in vitro pull-down experiments to evaluate the direct interaction between S-HDAg and BAZ2B BRD. To this end, the Strep-tagged ST-S-HDAg was expressed in Huh7 cells and purified by affinity chromatography using Strep-Tactin® magnetic beads. Sodium butyrate (5 mM) was added to preserve S-HDAg acetylation that is required for Pol II-mediated HDV genomic RNA synthesis and HDAg mRNA transcription[10,22,23]. The purity of the ST-S-HDAg protein was verified by silver staining (Supplementary Fig. 4a). StrepTactin® beads bind specifically to ST-S-HDAg and not to the untagged protein or to histone H3 (Supplementary Fig. 4b). Purified ST-S-HDAg was then incubated with bacterially expressed recombinant His6-BAZ2B BRD or His6-GFP (negative control). Immunoblot analysis of eluates from the affinity resin revealed the specific binding of His6-BAZ2B BRD to ST-S-HDAg (Fig. 2g, compare lanes 2 and 4) but not to His6-GFP (Fig. 2g, compare lanes 1 and 3). Furthermore, ST-S-HDAg purified from Huh7 cells using StrepTactin® magnetic beads and subsequently eluted by biotin was specifically retained on a BAZ2B BRD-linked Nickel-Nitrilotriacetic acid (Ni-NTA) agarose resin (Supplementary Fig. 4c). Altogether, these results indicate that S-HDAg recruits BRF complexes onto the viral RNP by directly interacting with the bromodomain of the regulatory subunit BAZ2B.

**S-HDAg contains a histone mimic recognized by BAZ2B BRD**. Structural and biophysical analysis of BAZ2B BRD complexes with histone-derived peptides have revealed that the interaction occurs through a short linear motif (SLiM) present in both H3 and H4 tails (K14acAPR in H3 and K16acRHR in H4 (Fig. 3a)[24]. This interaction requires the presence of an acetylated lysine in position 1 of the KacXXR SLiM, as well as an arginine at SLiM position 4 (R17 in H3 and R19 in H4) (Fig. 3)[24]. The second and third residues of the SLiM motif tolerate wide amino-acid changes with a strong preference for hydrophobic or aromatic residues at SLIM position 3[24]. The HAND domain of the *Drosophila* ISWI ATPase is acetylated at K753 in vitro and in live cells by GCN5[25]. This acetylated lysine is conserved in the mammalian ISWI orthologs SNF2L (K814) and SNF2H (K799) proteins, followed by a VPR sequence, thus generating a potential BAZ2B BRD SLiM (Fig. 3a). The alignment of 274 S-HDAg sequences showed a perfectly conserved SLiM motif across all eight genotypes with both K72 (SLiM position 1) and R75 (SLiM position 4) amino-acid residues conserved in all isolates from the eight HDV clades[26,27]. The KacVPR motif in SNF2L/H and the KacR/KA/PR motif in S-HDAg mimic the H3 and H4 KacXXR motifs and represent good candidates as BAZ2B BRD-binding sites[24]. We thus hypothesized that S-HDAg acetylation mediates BRF chromatin remodeler recruitment on the viral RNA replication complex. To validate the role of the S-HDAg SLiM R75 residue in the interaction between S-HDAg and BAZ2B BRD, we generated two Huh7 cell lines that stably express wt or R75A S-HDAg. Immunofluorescence staining and subcellular fractionation experiments confirmed the nuclear localization of R75A S-HDAg (Fig. 3b, c). Wild-type and R75A S-HDAg proteins showed similar levels of acetylation (Fig. 3d) and comparable half-life/protein stability (Fig. 3e). When cells were transfected with a GFP-Tag-BAZ2B BRD expression vector, BAZ2B BRD co-precipitated with acetylated histone H3 and with S-HDAg, in cells expressing wt S-HDAg (Fig. 3f). In contrast, a decrease in co-precipitation occured in cells expressing the R75A S-HDAg (Fig. 3f). These results support the notion of S-HDAg K72acXXR75 sequence acting as a SLiM for the interaction with BAZ2B BRD and the recruitment of BRF complexes on the HDV RNP.

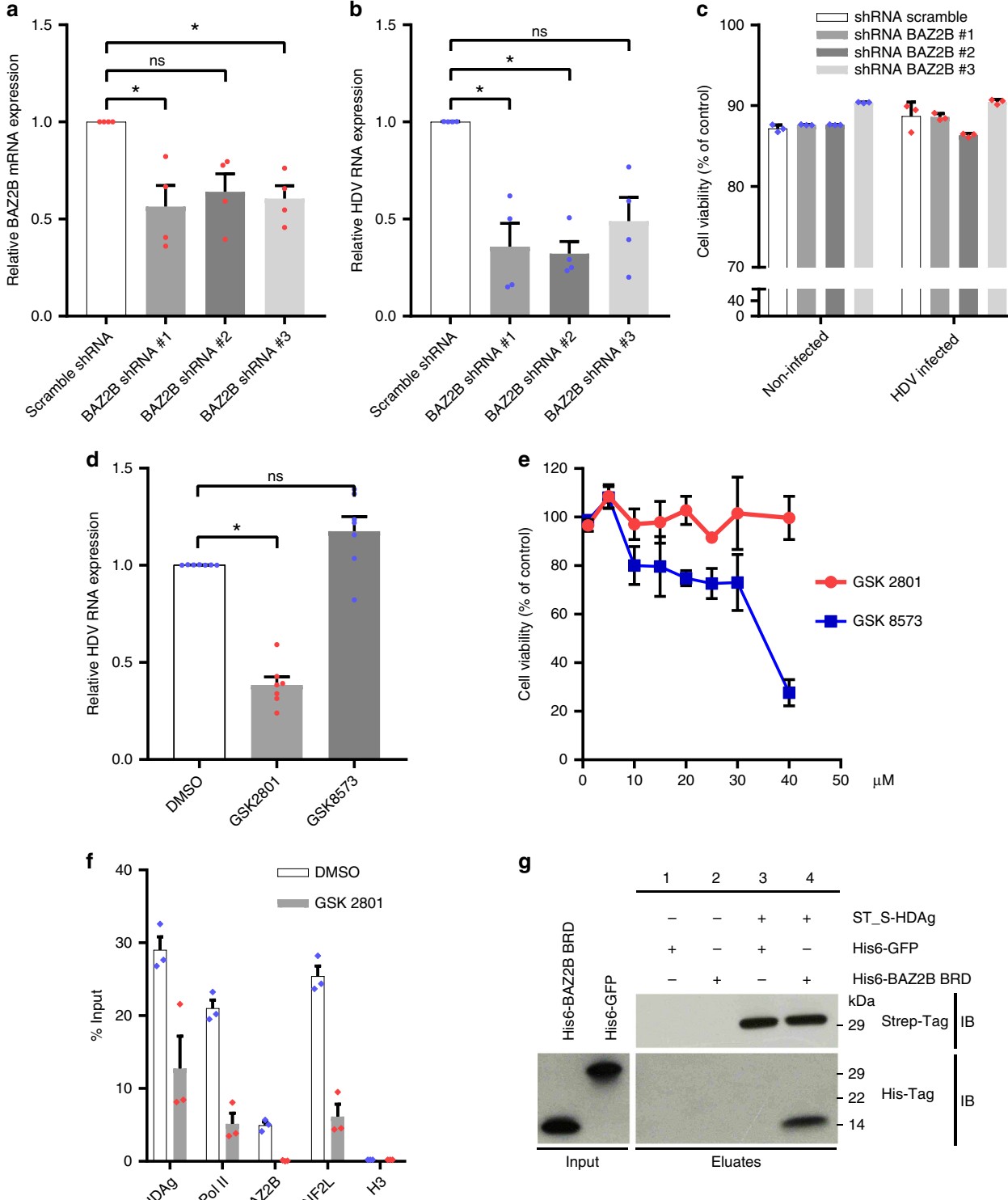

**R75A HDV is deficient for replication in infected PHHs**. To assess the effect of R75A substitution in S-HDAg on HDV RNA replication, we first co-transfected Huh7 cells with the replication-defective vector pSVL-D2M and either a wt or a R75A S-HDAg-expressing vector. As shown in Fig. 4a, R75A S-HDAg is defective in supporting HDV RNA replication, as evidenced by northern blotting analysis of intracellular genomic and anti-genomic HDV RNA (upper panels). Genomic HDV RNA levels were reduced by 65%, 50%, and 40% in R75A S-HDAg-transfected cells as compared with wt S-HDAg cells at 3, 6, and

9 days post transfection, respectively (Fig. 4a, lower left panel). A similar, albeit less pronounced, effect was observed when HDV antigenomic RNA levels were analyzed (Fig. 4a, lower right panel).

To further assess the effect of S-HDAg R75A substitution on HDV RNA replication in the setting of a relevant HDV infection system, wt or R75A HDV virions were first produced in Huh7 cells by co-transfection of a recombinant 1.3 mer of the HDV genome encoding the wt or R75A S-HDAg together with an HBV envelope proteins expression vector (pT7HB2.7). Following

**Fig. 2 BAZ2B enhances HDV replication and its BRD interacts with S-HDAg. a–c** Decrease in HDV RNA replication in PHHs transduced with shRNA against BAZ2B. PHHs were transduced separately with lentiviral vectors expressing shRNAs against BAZ2B (shRNA BAZ2B#1/shRNA BAZ2B#2/shRNA BAZ2B#3) or scramble shRNAs. PHHs were infected 72 h post transduction with HDV (m.o.i = 10). Relative BAZ2B mRNA (**a**) and HDV RNA expression (**b**) levels were analyzed by RT-qPCR 12 days post transduction. Results were normalized to GAPDH mRNA and represented as relative levels of mRNA in comparison to scramble shRNA. Values represent the mean ± SEM ($n = 4$); *$P < 0.05$; ns nonsignificant (Kruskal–Wallis test). **c** PHHs survival was assessed by the red neutral assay following BAZ2B shRNAs transduction in non-infected and HDV-infected cells. Values represent the mean ± SEM ($n = 3$). **d** HDV RNA replication is decreased in PHHs treated with the BAZ2B inhibitor GSK2801. PHH were treated with GSK2801, GSK8573, or DMSO. PHHs were infected with HDV (m.o.i = 10) and maintained in the presence of the inhibitor or DMSO until cell collection. Relative HDV RNA expression levels were analyzed by RT-qPCR 8 days post infection. Results were analyzed as in **b**. Values represent the mean ± SEM ($n = 7$); *$P < 0.05$; ns nonsignificant (Kruskal–Wallis test). **e** PHHs survival was assessed as in **c** following treatment with either GSK2801 or GSK8573. Values represent the mean ± SEM ($n = 3$). **f** HDV RIP in HDV-infected PHHs treated with GSK2801. PHHs were treated as described in **d** with either GSK2801 or DMSO. HDV RNA was immunoprecipitated with antibodies directed against HDAg, RNA Pol II, BAZ2B, SNF2L, and Histone H3. HDV RNA was detected by qRT-PCR using HDV-specific primers. Values represent the mean ± SEM ($n = 3$). **g** Pull-down assay of Strep-Tagged S-HDAg and BAZ2B BRD. Strep-Tactin® magnetic beads bound ST S-HDAg was mixed with his-tag BAZ2B BRD (5 µM) or his-tag GFP (5 µM). Input corresponds to 0.2 µM of recombinant protein and 30% of the SDS eluate was subjected to immunoblotting with anti-His-Tag antibody, whereas 5% of the SDS eluate was subjected to immunoblotting with anti-Strep-Tag antibody. Source data are provided as a Source Data file.

transfection, the kinetics of virion production was monitored by detection of HDV RNA in the culture medium and the kinetics of HBsAg levels were monitored by enzyme-linked immunosorbent assay to control transfection efficiency. Although the levels of HBsAg were similar in cells transfected with wt or R75A HDV cDNAs, northern blotting analysis (Supplementary Fig. 5a) showed that R75A virions were produced in lesser amounts, as compared with the wt. This is in agreement with R75A HDV RNA replication deficiency in transfected cells. Preparations of wt and R75A HDV virions were then normalized based on HDV RNA titers before inoculation of susceptible PHHs cultures at a multiplicity of infection (m.o.i.) of 10 (Supplementary Fig. 5b). Infection was monitored by measurement of intracellular HDV RNA at 2, 4, 6, and 8 days post inoculation (dpi). As shown in Fig. 4b, at any time point post inoculation, the levels of HDV RNA in R75A HDV-infected cells were at least 1.5 log lower than that of wt-infected cells. When infection was monitored for the presence of intracellular HDAg protein, a similar difference was observed between R75A- and wt-infected cells (Fig. 4b, upper panels). Additional infection experiments conducted in the NTCP-expressing Huh-106 cell line led to identical results (Supplementary Fig. 5c, d). Notably, in PHHs infected with the R75A virus, the recruitment of both Pol II and BRF proteins onto the viral RNP is severely impaired, as compared with wt HDV (Fig. 4c). Altogether, these results confirm the role of the S-HDAg R75 residue for the interaction with BRF complexes and, although we cannot exclude that the R to A substitution might affect other functions of the S-HDAg protein, they implicate this interaction in HDV replication.

## Discussion

The acetylation of the S-HDAg protein plays a crucial role in the synthesis of HDV RNA by cellular Pol II[22]. S-HDAg acetylation by the histone acetyl transferase p300 in the nucleus is mediated by the transcription factor YY1 and the interaction with a multiprotein complex of high molecular weight[22]. Here we show that the S-HDAg acetylation motif K72acXXR mimics the histone KacXXR SLiM and is read by the BRD of the BAZ2B protein, the regulatory subunit of the BRF chromatin remodeling complexes[18]. BAZ2B was identified by MS as one of the top 30 proteins that co-precipitate with the SNF2L and/or SNF2H ATPases[18]. Although both ATPases confer nucleosome-sliding activity to the BRF-1 and the BRF-5 complexes, respectively, an alternative splicing can generate a catalytically inactive SNF2L (ex13) variant in humans[19]. We found that both BRF-1/5 complexes containing active SNF2L/H ATPase subunits and BRF-1 complexes containing the catalytic inactive SNF2L(ex13) subunit

are present in human hepatocytes. The results of the RIP experiments clearly demonstrate that BRF remodelers are associated with HDV RNP in HDV RNA-infected cells. Experiments conducted both in vitro and in HDV-infected cells identify the crucial role of R75 at position 4 of the S-HDAg SLiM in its interaction with BRF remodelers. An HDV carrying the R75A mutation is highly impaired in the recruitment of the BRFs on the pseudo-double-stranded HDV RNA and displays reduced level of HDV RNA replication. Although we cannot exclude that the R to A substitution might affect other functions of the S-HDAg protein, our results support a model in which, by acting as a histone mimic, the acetylated SLiM-like sequence in S-HDAg confers the capacity to recruit the cell RNA Pol II and the associated chromatin remodelers onto the viral RNA (Fig. 5).

BAZ2B BRD has been shown to bind multiple acetylated lysine residues in histone peptide microarrays (histones H1.4, H2A, H2B, H3, and H4) with the highest binding affinity recorded in solution by isothermal calorimetry to histone H3 K14ac[28]. As the H3K14 acetylation mark is well represented in the 1 kb region downstream from the transcription start site in gene promoters[29], one would expect that this particular region is preferably targeted by BRF complexes to favor transcription initiation and/or elongation. Similarly, the HDV K72AcRAR SLiM might recruit BRF complexes onto the viral RNP to stimulate both transcription initiation and elongation in the rolling-circle amplification of HDV genome. The recruitment of BRF-1 complexes containing the catalytic-dead isoform SNF2L(ex13)[19] present in liver cells, if confirmed, would suggest that the recruitment of BRF complexes might have additional functions, beyond remodeling of the HDV RNP, which might have an impact on HDV RNA replication. HDV RNA is known to associate with the pre-mRNA splicing factors p54nrb and PSF, which are the two core protein components of the mammalian paraspeckles[30]. The knockdown of p54nrb, PSF, or PSP1 paraspeckle proteins reduces HDV RNA genome accumulation[31]. Recent evidence links components of mammalian SWI/SNF complexes, namely BRG1 and BRM, to paraspeckle assembly without requiring their ATPase activity[32]. Thus, it is tempting to speculate that recruitment of catalytic-dead BRF-1 might serve to direct the transcriptionally active HDV RNP to paraspeckles.

Histone mimicry is used by certain viruses to subvert normal host defenses[33]. For instance, the influenza non-structural protein 1 from the H3N2 subtype uses the histone H3K4-like sequence 226ARSK229 at its carboxyl terminus to bind to and hijack the human PAF1 transcription elongation complex (HPAF1C), thus affecting the expression of antiviral genes[34]. The histone H4-like protein CpBV-H4 of *Cotesia plutellae* bracovirus competes with

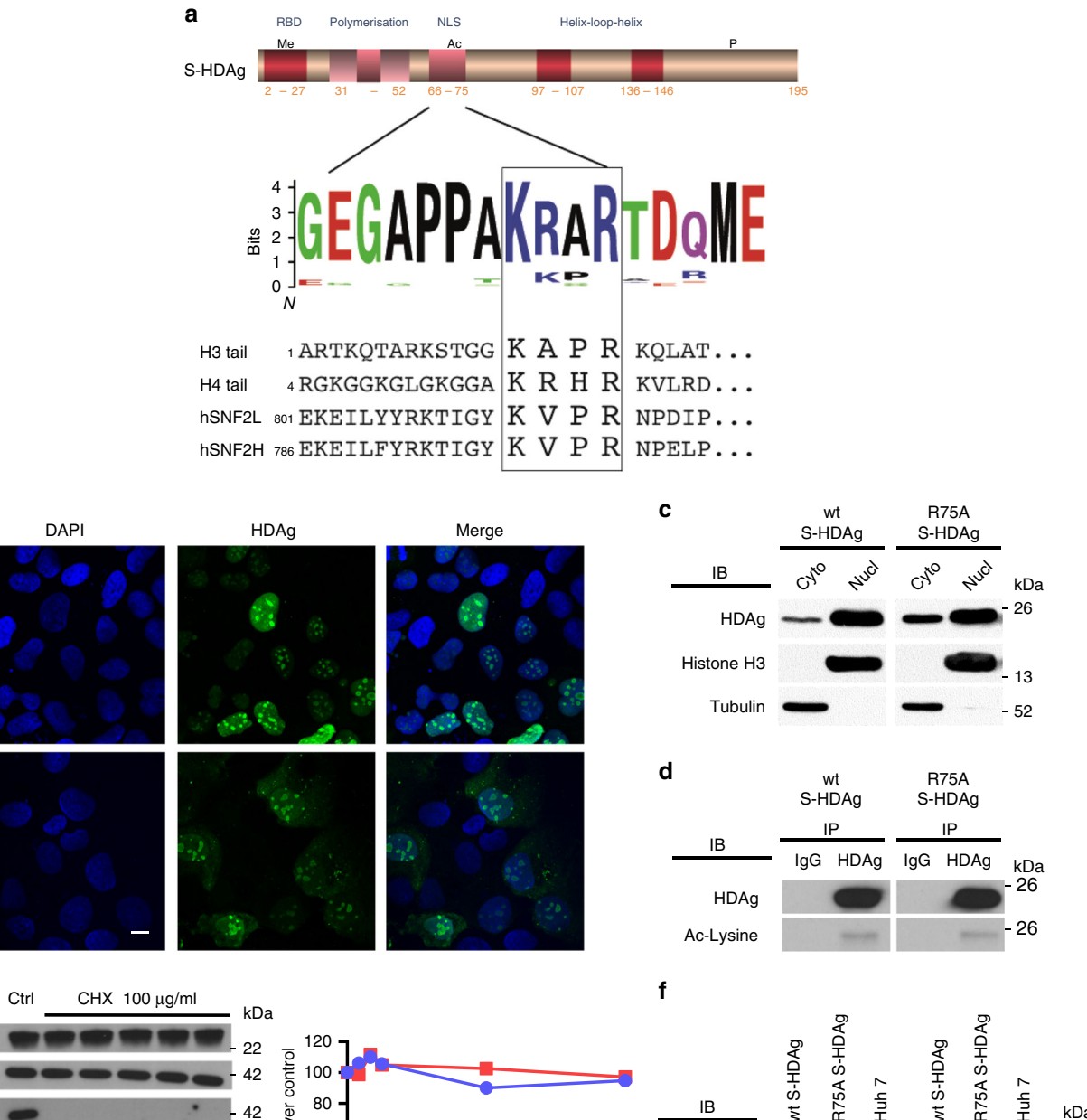

endogenous cellular H4 for incorporation into chromatin and alters host immune-associated genes expression[35]. The adenovirus protein VII, which resembles cellular histones, complexes with nucleosomes to sequester the high-mobility-group protein B1, making it unable to signal danger and activate immune responses[36]. Our results identify a new role of histone mimicry. HDV exploits histone mimicry to confer a novel function to a viral protein and to compensate the lack of a viral replicase. The acetylated SLiM-like sequence in S-HDAg empowers the virus to hijack the host transcription co-factors necessary for the replication of its genome (Fig. 5).

HDV infections represent a major health problem worldwide. Fast disease progression and lack of an effective treatment result

in an increased risk of end-stage liver complications including cirrhosis, liver failure, hepatocellular carcinoma, or death. Pegylated interferon, the only authorized treatment for HDV-HBV co-infected chronic patients, may control the disease but has limited long-term effects. New HDV therapies are actively investigated. Indeed, the unique HDV life cycle and its dependence on HBV for its entry and egress are exploited for antiviral intervention. The HBV entry inhibitor Myrcludex B®, the HBsAg secretion inhibitors REP2139® and REP2165®, and the viral assembly and release inhibitor Lonafarnib, which are in early clinical development, all interfere with HBV envelope functions or the interaction between HDAg and the HBV envelope[37]. Direct inhibition of HDV RNA replication has not been achieved

**Fig. 3 R75A S-HDAg mutation affects the binding to BAZ2B BRD without altering S-HDAg localization and acetylation. a** Top panel: schematic representation of S-HDAg domains. HLH helix loop helix domain encompassing arginine-rich motifs (ARM), LZ leucine zipper-like polymerization domain, NLS nuclear localization signal, RBD RNA-binding domain. Middle panel: consensus alignment of 274 HDAg sequences displayed as a WebLogo® showing the K72ac-X-X-R75 motif (squared) with perfect conservation of the K72 and R75 residues. Bottom panel: alignment of H3 and H4 tails, hSNF2L and hSNF2H indicating the acetylated motif for each protein. **b** Wt S-HDAg and R75A S-HDAg proteins have a similar nuclear localization pattern. Huh7 cells, transfected with plasmids coding for either wt or R75A S-HDAg proteins, were subjected to indirect immunofluorescence at day 3, using an HDAg antibody (green), and nuclei were stained using DAPI (blue). Images were captured by confocal microscopy (objective ×63; digital zoom 0.8; bar = 10 μm). **c** Wt and R75A S-HDAg are expressed at the same level. Huh7 cells stably expressing wt or R75A S-HDAg proteins were subjected to subcellular fractionation. Wt and R75A S-HDAg levels in the nuclear (Nucl) and cytoplasmic (Cyto) fractions were determined by IB using the α-HDAg antibody. The α-Alpha Tubulin and α-Histone H3 antibodies verified fraction purity and loading amount. **d** Wt and R75A S-HDAg have similar acetylation levels in Huh7 cells stably expressing each protein. Equal quantities of nuclear protein extracts were immunoprecipitated with α-HDAg and immunoblotted with antibodies against acetyl-lysine. **e** Wt S-HDAg and R75A S-HDAg protein stability was compared in Huh7 cells treated with cycloheximide (CHX: 100 μg/ml). Left panel: Total cell extracts were analyzed by western blotting using the indicated antibodies. Right panel: Densitometric values expressed as ratio over wt S-HDAg or R75A S-HDAg controls (time 0). **f** Co-immunoprecipitation of S-HDAg and GFP-Tag-BAZ2B BRD. Parental Huh7 cells and Huh7 cells stably expressing wt or R75A S-HDAg were transfected with the pGFP-Tag-BAZ2B BRD plasmid coding for a GFP-BAZ2B BRD fusion protein targeted to the nucleus. Nuclear extracts were subjected to IP with GFP-Trap® beads. Input and elutions were analyzed by IB using the indicated antibodies. Source data are provided as a Source Data file.

so far. The recruitment of BRF complexes onto the HDV RNP and the role of S-HDAg interaction with the BAZ2B BRD in HDV replication underlines the involvement of additional cellular factors, besides the Pol II and its partners in the host basal transcriptional machinery, and may help to identify new potentially druggable targets for HDV.

## Methods

**Cell culture**. The HepaRG cells were a kind gift from Philippe Gripon[38] and can be purchased from Biopredict International. HepaRG cells and the HepaRG ST S-HDAg cell line conditionally expressing a double Strep-Tagged recombinant S-HDAg were cultured at 37 °C in a humidified atmosphere containing 5% $CO_2$ in Williams E medium (Life Technologies) supplemented with 10% Hyclone Fetal clone II serum (Thermo Scientific), 2 mM Glutamax (Life Technologies), 5 μg/ml insulin, $7 \times 10^{-5}$ M hydrocortisone hemisuccinate, and 50 U/ml of antibiotics Penicillin/Streptomycin (Pen Strep, Life Technologies). To induce differentiation, HepaRG cells were cultured for 2 weeks until confluence in Williams complete medium subsequently supplemented with 1.8% dimethyl sulfoxide (DMSO) (Sigma Aldrich) for 2 more weeks. The human hepatoma Huh7 cell line was a kind gift from Christoph Seeger and is described in Durantel et al.[39]. Huh7 cells and Huh7 cells lines stably expressing S-HDAg wt and R75A mutant S-HDAg protein were cultured at 37 °C in a humidified atmosphere containing 5% $CO_2$ in Williams E medium (Life Technologies) supplemented with 10% Hyclone Fetal clone II serum (Thermo Scientific), 2 mM Glutamax (Life Technologies), 25 mM Hepes, and antibiotics 10 μg/ml Gentamicin (Life Technologies). The HEK293T cell line was obtained from the ATCC (ATCC® CRL-3216™) and were cultured at 37 °C in a humidified atmosphere containing 5% $CO_2$ in Dulbecco's modified Eagle medium supplemented with 10% fetal bovine serum and 50 U/ml of antibiotics Penicillin/Streptomycin (Pen Strep, Life Technologies).

For PHHs isolation, HBV, HCV, and HIV negative human liver resections were provided by Michel Rivoire from the Centre Léon Bérard, Lyon, under the French ministerial authorizations (AC 2013–1871, DC 2013–1870, AFNOR NF 96 900). Prewashed liver resections underwent a standardized two-step collagenase perfusion[40]. PHHs were seeded on rat-tail collagen (0.2 mg/ml, Corning® Collagen I)-coated plates at a cell density of 20E5 cells/cm² and cultured at 37 °C in a humidified atmosphere containing 5% $CO_2$ in serum-free William's E medium including insulin 5 mg/mL (Sigma Aldrich), penicillin 100 IU/mL, streptomycin 100 mg/mL (Life Technologies), and hydrocortisone hemisuccinate (50 mM, Pharmacia Upjohn). PHH culture medium was changed 12–16 h after extensive washing with phosphate-buffered saline (PBS). Cell viability and number were verified by Trypan blue exclusion method and hepatocyte preparations with a viability exceeding 85% were chosen for subsequent experiments. Culture medium was additionally supplemented with 5% Hyclone Fetal clone II serum (Thermo Scientific) and 2% DMSO (Sigma Aldrich), and was replaced with fresh medium every 2 days.

**Generation of stable cell lines**. The HepaRG ST-S-HDAg cell line was established by transducing HepaRG cells with lentiviruses generated upon the co-transfection of HEK293T cells with the lentiviral vector pSLIK[41] (Addgene, Cat#: 25737) and packaging vectors pHCMV-G[42] (Cat#: ATCC® 75497™) and pCMVΔ8.2 HIV-1 Gag-Pol[43] (Addgene, Cat#: 12263). HEK293T cells were co-transfected with calcium phosphate precipitation and cell media was replaced 12 h post transfection. Supernatants collected 48 h post transfection were filtered (pore size 0.2 μm) and concentrated with an Amicon® Ultra 15 ml Centrifugal Filter at $4000 \times g$. Supernatants were premixed with polybrene (8 μg/ml, Millipore and 50 ng/ml

Epidermal Growth Factor (EGF) (Preprotech) and were used to infect proliferating HepaRG cells followed by Hygromycin B selection (100 mg/mL, Sigma Aldrich) 48 h post infection for the generation of a polyclonal cell population. The HepaRG ST S-HDAg cell expresses the Strep-Tag®-fused S-HDAg upon addition of Doxycycline (Sigma Aldrich). The Huh7 S-HDAg wt and S-HDAg R75A cell line were established by transfecting Huh7 cells with the pEXPR-S-HDAg wt cDNA or pEXPR-S-HDAg R75A cDNA plasmids, respectively, using the TransIT®-LT1 Transfection reagent (Mirus). Cell medium was replaced 24 h post transfection followed by G418 neomycin (500 μg/ml, Gibco BRL) selection. G418-resistant colonies were collected 20 days of selection. S-HDAg and S-HDAg R75A proteins were analyzed by western blotting and were subsequently chosen for further expansion in G418 (200 μg/ml)-supplemented medium.

**Virus production and HDV infections**. For production of HDV particles, Huh7 cells were co-transfected with the pSVLD3 plasmid for production of HDV RNPs and pT7HB2.7 for the expression of HBV env proteins[44]. Culture medium was collected 5, 7, and 9 days post transfection and analyzed for the presence of HDV viral by northern blotting hybridization for the detection of HDV RNA in the cell culture supernatant as described below. The wt or R75A HDV virions were produced in Huh7 cells by co-transfection of a recombinant 1.3 mer of the HDV genome encoding either the wt or R75A S-HDAg and the HBV envelope protein expression vector pT7HB2.7. Cell culture supernatants were collected 3–12 days post transfection. For in vitro infection assays, supernatant containing wt and mutant HDV virions were adjusted to 1.00E + 09 ge/ml. For infection assays, PHHs were inoculated with HDV at a m.o.i. of 10 ge/cell on day 1 post seeding in the presence of 5% PEG 8000 (Sigma) and were incubated at 37 °C for 16 h. Inoculum was removed and replaced with PHH medium, changing every 2 days[45]. Cells were collected at 8 dpi for measurement of intracellular HDV RNA.

**Plasmids**. The pGFP-Tag-BAZ BRD plasmid (kindly provided by Susanne Müller) encodes the green fluorescent protein (GFP) protein fused to the PHD-BRD containing a fragment derived from the C-terminus of BAZ2B and the SV40 nuclear localization signal. The plasmids pEXPR-S-HDAg and pEXPR-S-HDAg R75A express the wt S-HDAg protein and the R75A mutant S-HDAg. The plasmid pEXPR-S-HDAg wt was generated by inserting a 615-nucleotide synthetic cDNA fragment encompassing the S-HDAg genotype 1 coding sequence between the Xba1 and Not1 sites of the pEXPR-IBA105 (IBA, Cat#: 2–3505–000) empty vector. Similarly, the plasmid pEXPR-S-HDAg R75A contains the same insert including the R to A mutation at codon 75 (Eurofins Scientific). Plasmids pSVL-D2M[4] and pSVLD3 were kindly provided by John M. Taylor (Fox Chase Cancer Center, USA). All plasmid constructs were sequence verified.

**Cell transfection**. HepaRG or Huh7 cells were seeded at 60–80% confluency and were allowed to adhere overnight. Cells were then transfected with indicated amounts of total plasmid DNA, in the presence of TransIT®–LT1 in serum-free Opti-MEM medium (Life Technologies) following the manufacturer's instructions. After 24 h, the transfection medium was removed and was replaced with fresh medium.

**Immunofluorescence**. Cells cultivated on glass coverslips were fixed with 2% paraformaldehyde in PBS buffer for 5 min and next permeabilized with 0.3% Triton X-100 in PBS for 10 min. After a blocking step with 3% bovine serum albumin (BSA) in PBS 1× buffer, cells were incubated with an anti-HDAg polyclonal antibody (in-house rabbit antibody 1:50 dilution[46]) for 1 h at room temperature. Secondary labeling was performed by incubating coverslips with Alexa Fluor fluorescent antibodies (Thermo Scientific, Cat#: A-11034, RRID:

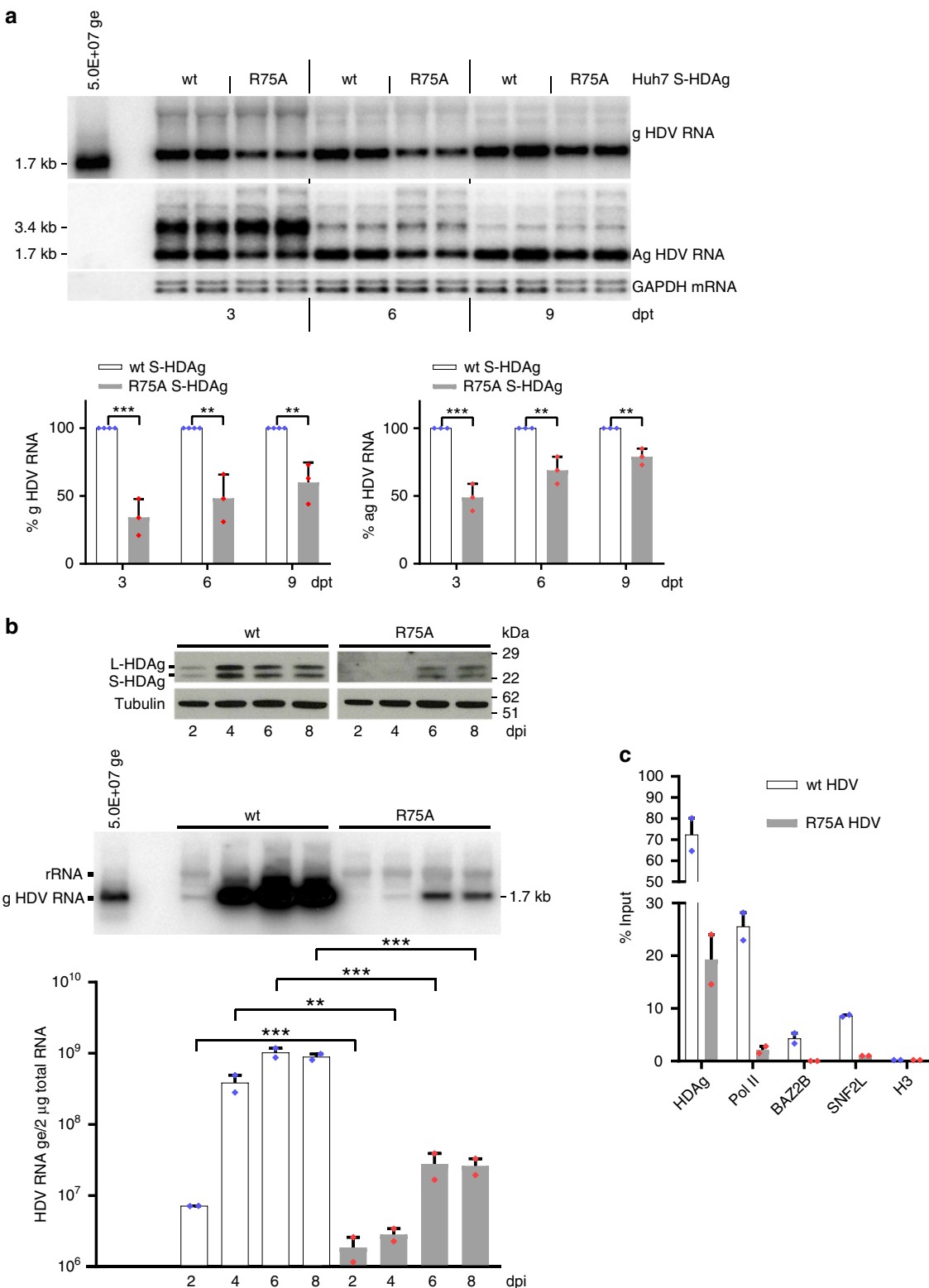

AB_2576217 or Cat#: A-21070, RRID: AB_2535731, dilution 1:1000) diluted in PBS/BSA. Nuclei were counter stained with 4.6-diamidino-2-phenylindole or Hoechst. All images were acquired by epifluorescence microscopy (Nikon eclipse TE2000–E; Nikon).

**Total cell protein extract and cellular fractionation**. Cells were collected with Trypsin-EDTA and centrifuged at $800 \times g$ for 8 min at 20 °C. Cell pellets were lysed in the Mammalian Protein Extraction Reagent (M–PER; Pierce) supplemented with 2 mM phenylmethanesulfonyl fluoride (PMSF), cOmplete™ Mini EDTA-free Protease Inhibitor Cocktail (Roche), 1 U/ml Benzonase® nuclease (Sigma Aldrich),

and 1 mM $MgCl_2$. Cytoplasmic and nuclear fractionation was carried out by using the NE–PER™ kit (Thermo Scientific) as per the manufacturer's instructions.

**Western blotting and antibodies**. Cell protein extracts were incubated in Laemmli buffer (62 mM Tris-HCl pH 6.8, 2% SDS, 10% glycerol, 100 mM dithiothreitol (DTT), and 0.01% phenol red) at 95 °C for 5 min. Protein concentrations were determined with BCA protein assay kit (Thermo Scientific). Protein lysates were separated on 12% SDS polyacrylamide gels or 4–20% Tris-HCl Criterion™ Precast gels (Bio-Rad) and resolved proteins were electro-transferred onto nitrocellulose membranes (Bio-Rad). After saturation with 5% low-fat milk

**Fig. 4 Loss of HDAg binding to BAZ2B reduces HDV replication and L-HDAg expression. a** Representative northern blotting indicating the decrease in the gHDV RNA accumulation in the presence of the R75A S-HDAg protein. Huh7 cells stably expressing wt or R75A S-HDAg were transfected with the pSVL-D2M plasmid expressing a S-HDAg-defective HDV RNA dimer. Total RNAs were extracted at the indicated times post transfection (dpt) and analyzed by northern blot assay (upper panel). Each lane was loaded with 3 μg of total RNA and probed with a $^{32}$P-radiolabeled ag or gHDV riboprobes. Histograms represent signal intensities of the g or ag HDV RNA accumulation in the presence of R75A S-HDAg (grey bars) relative to signals measured in the presence of the wt S-HDAg protein (black bars) (lower panels). Band densities were quantified using ImageJ software and normalized using the GAPDH mRNA. Values represent the mean ± SEM ($n = 3$); **$P < 0.005$; ***$P < 0.0005$ (one-way ANOVA). **b** PHHs infection with wt HDV or recombinant R75A HDV mutant. Viral particles were inoculated in PHHs at a m.o.i. of 10. Cells were collected at the indicated times post infection (dpi) and total cell protein extracts (upper panel) and total RNAs (middle-lower panels) were analyzed. Wt HDV (short exposure) and R75A HDV (long exposure) protein extracts were subjected to immunoblotting using human polyclonal anti-HDAg Abs (upper panel). Anti-tubulin antibody was used as a loading control. Northern blottings were performed on 5 μg total RNA with a radiolabeled antigenomic probe detecting HDV genome molecules (film exposure = 3 hrs) (middle panel). PhosphoImager quantification of unsaturated exposures (lower panel) ($n = 2$). Hybridization was calibrated using a 5.00E + 7 in vitro-transcribed genome signal. Band densities were quantified using ImageJ software and normalized using the 28s rRNA. Error bars indicate the SEM from two independent experiments. **c** HDV RIP in PHHs infected with wt HDV or recombinant R75A HDV mutant. HDV RNA was immunoprecipitated with antibodies directed against HDAg, RNA Pol II, BAZ2B, SNF2L, and Histone H3. HDV RNA was detected by qRT-PCR using HDV-specific primers. Values represent the mean ± SEM ($n = 2$); **$P < 0.005$ and ***$P < 0.0005$ (one-way ANOVA). Source data are provided as a Source Data file.

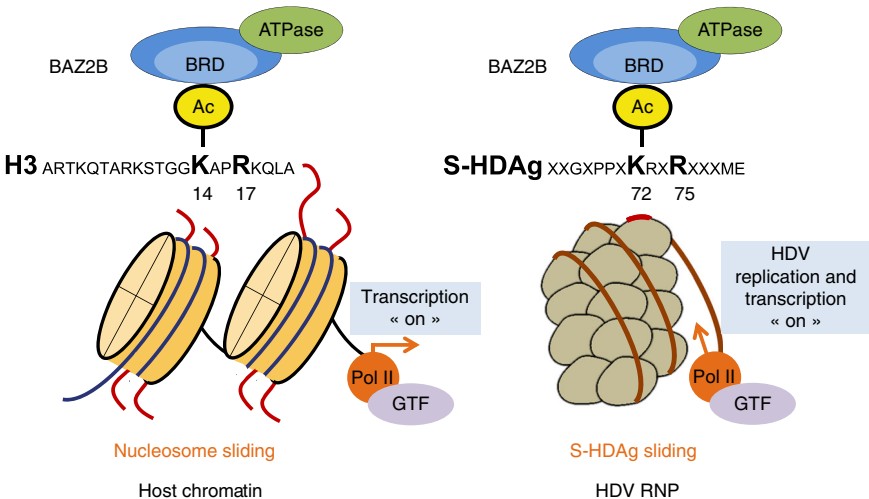

**Fig. 5 Modeling the role of S-HDAg acetylation and BRF chromatin remodelers in HDV viral replication.** The K72acXXR75 sequence in the S-HDAg protein mimics histone H3 K14acXXR17 motif and interacts with the bromodomain (BRD) of BAZ2B. The recruitment the BRF chromatin remodelers to the HDV RNP empowers the virus to hijack the host transcriptional machinery for the replication of its RNA genome. The proposed model is based upon the RIP experiments shown in Figs. 2c, 2d, and 4c; the in vitro pull-down assays shown in Fig. 2f and Supplementary Fig. 4c; the co-immunoprecipitation experiments shown in Fig. 3f and the PHHs infection experiments with wt HDV and R75A recombinant HDV shown in Fig. 4b.

powder in PBS with 0.05% Tween-20 for 1 h at room temperature, membranes were incubated with primary antibodies overnight at 4 °C. The following primary antibodies were used: (a) Mouse monoclonal antibodies: Anti-Strep-Tag® (Qiagen, Cat#: 34850, 1:1000 dilution), Anti-Tubulin (Santa Cruz, Cat#: sc-8035, RRID: AB_628408, 1:5000 dilution), Anti-GFP (Abcam, Cat#: ab38689, RRID: AB_732715, 1:5000 dilution), Anti-β-Actin (Cell Signaling, Cat#: 3700, RRID: AB_2242334, 1 :5000 dilution), Anti-MCL1 (Abcam, Cat#: ab32087, RRID: AB_776245, 1:1000 dilution); Rabbit polyclonals: Anti-HDAg (in-house antibody produced by Dr Alan Kay, 7.1 μg/ml), Anti-Lamin-B1 (Abcam, Cat#: ab16048, RRID:AB_443298, 1:1000 dilution), Anti-Histone H3.3 (Diagenode, Cat#: C15210011, 1:1000 dilution), Anti-Histone H3 acetyl K9/K14/K18/K23/K27 (Abcam, Cat#: ab47915, RRID:AB_873860, 1 :1000 dilution), Anti-H3K27me3 (Abcam, Cat#: ab6002, RRID:AB_305237, 1 :1000 dilution), Anti-SNF2L (Abcam, Cat#: ab37003, RRID:AB_945532, 1:500 dilution), Anti-SNF2H (Abcam, Cat#: ab3749, RRID:AB_2191856, 1:500 dilution), and Anti-His-tag (Invitrogen, Cat#: MA1-135, RRID:AB_2536841, 1:1000 dilution). Horseradish peroxidase-linked secondary antibodies (Mouse [Cell Signaling, Cat#: 7076, RRID:AB_330924, 1:10000 dilution], Rabbit [Cell Signaling, Cat#: 7074, RRID:AB_2099233, 1:10,000 dilution]), Human (Abcam, Cat#: 6759, RRID:AB_955434, 1:5000 dilution) were detected by chemiluminescence (SuperSignal™West Pico Chemiluminescent Substrate, Thermo scientific). Uncropped and unprocessed scans of all relevant blots are included in the Source Data file.

**Co-immunoprecipitation**. For co-immunoprecipitation assays cell media were supplemented with 5 mM Sodium Butyrate 24 h prior collection. RIP was performed using $10^7$ cells/antibody condition. Cell nuclei were isolated using the Nuclei Isolation Buffer (NIB: 0.256 M Sucrose, 8 mM Tris-HCl pH 7.5, 4 mM MgCl₂, 1% Triton X-100, cOmplete™ Mini Protease Inhibitor Cocktail [Roche], 1 mM PMSF) supplemented with 20 mM Sodium Butyrate for 30 min on ice followed by dounce homogenization. After centrifugation at $2500 \times g$ for 15 min at 4 °C, the supernatants were discarded and the nuclei pellet was washed with PBS supplemented with cOmplete™ Mini Protease Inhibitor Cocktail (Roche), PMSF (1 mM), and cleared by centrifugation at $2500 \times g$ for 15 min at 4 °C. Nuclei enriched pellet was further lysed in RIPA buffer (150 mM KCl, 25 mM Tris-HCl pH 7.4, 5 mM EDTA, 0.5 M DTT, 0.5% NP40, cOmplete™ Mini Protease Inhibitor Cocktail [Roche], PMSF) supplemented with Benzonase® nuclease (Sigma Aldrich, final concentration 1 U/ml), 20 mM Sodium Butyrate, and 1 mM MgCl₂ on ice for 40 min. Nuclei lysates were cleared by centrifugation at $13,000 \times g$ for 10 min at 4 °C. Supernatants were pre-cleared by incubation with Protein G-sepharose beads (GE Healthcare) for 2 h at 4 °C. To couple the antibodies to the Dynabeads™ Protein G (Invitrogen), indicated antibodies (Anti-HDAg antibody-positive human serum, Anti-BAZ2B [Life technologies, Cat#: 730006, RRID:AB_2532814, final concentration 30 μg/ml], Anti-SNF2L [Abcam, Cat#: ab37003, RRID:AB_945532, final concentration 10 μg/ml], Anti-SNF2H [Abcam, Cat#: ab3749, RRID: AB_2191856, 10 μg/ml], or nonspecific control Rabbit antibody IgG [Abcam, Cat#: ab171870, RRID:AB_2687657, 20 μg/ml]), were added to 50 μl of beads resuspended in 0.5 ml of Antibody Coupling Buffer (ACB: 3% BSA, cOmplete™ Mini Protease Inhibitor Cocktail [Roche], 1 mM PMSF) overnight at 4 °C, one day prior to the RIP assay. To remove the uncoupled antibodies, beads were washed thrice with ACB and equilibrated in 100 μl of ACB. Pre-cleared nuclear lysates were next incubated overnight at 4 °C with the beads coupled to the indicated antibodies. After thorough washing with RIPA buffer, beads were resuspended in Laemmli buffer (2×) and incubated at 95 °C for 10 min. Immunoprecipitated samples were resolved using 4–20% precast SDS polyacrylamide gels. Horseradish peroxidase-linked secondary antibodies (Veriblot [Abcam, Cat#: 131366, 1:1000 dilution])

were detected by chemiluminescence. For co-immunoprecipitation assays involving the GFP-TRAP®_M beads (Chromotek), the nuclei lysates were obtained as described above and incubated with 25 μl of GFP-TRAP®_M beads overnight at 4 °C. After thorough washing with RIPA buffer, beads were resuspended in Laemmli (2×) and incubated at 95 °C for 10 min prior to electrophoresis.

**shRNA-mediated silencing/shRNA interference for BAZ2B**. Lentiviruses containing shRNA for BAZ2B were obtained from GenTarget, Inc., in the pLenti-H1-shRNA-puro lentivector. BAZ2B shRNA sequences were designed using the Gentarget's software to knockdown human BAZ2B gene (Locus ID 29994), for all its transcripts variants (NM_001289975.1; NM_001329857.1; NM_001329858.1; NM_013450.3). The target sequences of the shRNA used are: shRNA BAZ2B #1 (5′-GATAGTGATTCAGGCACATCA-3′), shRNA BAZ2B #2 (5′-ATCCTGTTG GCTTAAATCCAT-3′), shRNA BAZ2B #3 (5′-GTATGGAGAAGGGCATTAT CA-3′), shRNA Scramble (5′-GTCTCCAGGCGCAGTACATTT-3′). Lentiviruses were transduced into PHHs overnight in the presence of 10 ng/ml of EGF (Gibco, Cat#: PHG0311L) and 8 μg/ml of Polybrene (Sigma Aldrich, Cat#: TR-1003-G). Cell medium was refreshed 24 h post transduction. Knockdown efficiency of BAZ2B mRNA expression was analyzed by RT-qPCR 12 days post transduction.

**Analysis of HDAg proteins in HDV viral inoculum**. Equal fractions of equally calibrated virus preparations (HDV WT or HDV S-HDAg R75A) were precipitated with 90% ethanol in the presence of glycogen on ice for 3 h followed by centrifugation at 15,500 × g at 4 °C for 1 h. The pellet was washed twice with 75% ethanol. The pellet was air-dried and resuspended in Laemmli buffer heated at 95 °C for 5 min and subjected to electrophoresis on a 12% SDS polyacrylamide gel.

**Inhibitors treatment**. GSK2801 (Sigma Aldrich, Cat#: SML0768) and GSK8573 (Biorbyt, Cat#: orb3155460) were resuspended in DMSO and diluted in cell culture medium to a final concentration of 10 μM for treatments and 10–50 μM for cell viability assays, respectively. The same volume of DMSO was added to all inhibitor-treated or control samples within each experiment. PHHs were incubated with GSK2801 or GSK8573, or DMSO 1 day prior to HDV infection. Twenty-four hours post HDV infection, inhibitors (or DMSO) were added to the cell culture media and were maintained throughout the infection by replacing with fresh medium containing inhibitors (or DMSO) every 2 days. PHHs survival was assessed by the Neutral Red uptake assay[47]. Precipitated dye crystal free neutral red medium (40 μg/ml Neutral red medium prepared with neutral red dye and PBS) was added to shRNA/inhibitor-treated or non-treated cells and incubated for 2 h at 37 °C. Neutral red medium was discarded followed by a PBS wash. Neutral red destain solution (50% Ethanol, 49% deionized water, 1% glacial) was next added to cells and plates were rapidly shaken on a microtiter plate shaker for 15 min. The optical density of the neutral red extract was measured at 540 nm in a microtiter plate reader spectrophotometer. The relative percent cell viability was calculated by dividing the luminescence values of transduced cells by the mean DMSO-treated cells.

**RNA isolation and qRT-PCR with reverse transcription**. Total cellular RNA was isolated from cultured cells with the Trizol reagent (Invitrogen). After addition of Trizol, samples were heated at 50 °C for 10 min and subsequently processed as per the manufacturer's instructions. RNA was treated with RNAse-free Ambion™ DNAse I (Invitrogen, Cat#: AM2222) in the presence of 1 U/μl SUPERase• In™ RNase Inhibitor (Invitrogen, Cat#: AM2694). The concentration and purity of the RNA were verified using the Nanodrop spectrophotometer and ethidium bromide stained 18S/28S RNA bands were visualized using an ultraviolet *trans*-illuminator after agarose gel electrophoresis. Total RNA (1 μg) was reverse-transcribed using Superscript III (Invitrogen, Cat#: 18080044) according to the manufacturer's recommendations. Quantification of HDV cDNA was carried out by quantitative PCR (qPCR) with the PowerUp™ SYBR® Green Master mix (Applied Biosystems, Cat#: A25779) in the Roche LightCycler® 480 instrument. Each reaction was run in technical duplicate or triplicate. Quantification was performed by applying the ΔΔCt method[48] normalizing to the GAPDH or RPLP0 gene expression to correct variation between samples. The HDV (forward and reverse), GAPDH (forward and reverse), and RPLP0 (forward and reverse) primers are detailed in Supplementary Table 2. All oligonucleotide primers were manufactured by Eurofins, except for BAZ2B primers, which were purchased from Qiagen (PPH 18869B-200).

**Northern blotting and hybridization[45]**. Total cellular RNA denatured with Glyoxal (Ambion, Cat#: AM8551) was subjected to electrophoresis on a 1.2% agarose gel at 60 mV for 3 h. The RNA samples were transferred from the agarose gel to a Amersham Hybond-N + membrane (GE Healthcare, Cat#: RPN119B) by upward capillary transfer for 16 h at 4 °C. Once the transfer completed, the membrane was incubated at 80 °C for 2 h. The detection of the genomic HDV RNAs was conducted by using $^{32}$P-labeled probes from pcDNA3 HDV 1 × 1 containing one copy of the HDV genome plasmid digested with HindIII to obtain an antigenomic probe using the MEGAscript™ SP6 transcription Kit (Invitrogen, Cat#: AM1330). The detection of the antigenomic HDV RNAs was conducted by using $^{32}$P-labeled probes from pcDNA3 HDV 1 × 1 plasmid digested with XbaI to obtain a genomic probe using MEGAscript™ T7 trancription Kit (Invitrogen,

Cat#: AM1333). After a 2 h pre-hybridization at 65 °C, probes ($10^6$ c.p.m./ml) were added and incubated overnight on constant rotation. Membranes were washed twice with 1% SDS, 1 × SSC solution (0.15 M sodium chloride and 0.015 M sodium citrate pH 7.0) and twice again with 0.1% SDS, 0.1 × SSC. Membranes were next incubated in an exposure cassette for 2–4 h and/or overnight and were either exposed with the PhosphoImager and/or developed using a Kodak Biomax film.

**SNF2L splice variant PCR assay**. Total RNA was reverse-transcribed to cDNA using Superscript™ III reverse transcriptase (Invitrogen) as per the manufacturer's instructions. PCRs were performed by using specific primers[19] (Supplementary Table 1) and Q5® High-Fidelity DNA Polymerase (NEB, Cat#: M0491S). The SNF2L Set A (forward and reverse) primers, which detect both ATPase-dead SNF2L(ex13) and ATPase-active SNF2L and the SNF2L Set B (forward and reverse) primers, which detect only the ATPase-dead SNF2L(ex13), are detailed in Supplementary Table 1.

**HDV RNA immunoprecipitation assay**. RIP was performed using $10^7$ Huh7 cells or PHHs per antibody condition. Cells were crosslinked for 10 min with 0.5% glutaraldehyde in William E media followed by quenching with 0.125 M Glycine in PBS for 5 min. Henceforth, every buffer was supplemented with 0.1 U/μl SUPERase In™ RNAse inhibitor (Invitrogen). Crosslinked cells were rinsed thrice with PBS supplemented with cOmplete™ Mini Protease Inhibitor Cocktail (Roche) and PMSF (1 mM), scraped and pelleted at 4,000 × g for 5 min. Cell nuclei were isolated using the NIB buffer for 30 min on ice followed by dounce homogenization ($10^7$ cells/ml per antibody condition). After centrifugation at 2500 × g for 15 min at 4 °C, the supernatants were discarded. The nuclear pellets were washed once with PBS supplemented with cOmplete™ Mini Protease Inhibitor Cocktail (Roche), PMSF (1 mM) and cleared by centrifugation at 2500 × g for 15 min at 4 °C. Nuclear pellets were further lysed in RIPA buffer on ice for 10 min ($10^7$ nuclei in 500 μl of RIPA buffer per antibody condition). Nuclear lysates were sonicated (30 s on and 45 s off, 2 cycles) using a Diagenode Bioruptor® pico water bath sonicator. Sonicated nuclear lysates were centrifuged at 13,000 × g for 10 min at 4 °C. Input RNA corresponded to 1% (v/v) of the supernatant. To couple the antibodies to the Dynabeads™ Protein G (Invitrogen), indicated antibodies (diluted as in the Co-immunoprecipitation section) as well as anti-RNA Pol II CTD phospho 5 (Abcam, Cat#: 5131, RRID:AB_449369, final concentration 10 μg/ml) or nonspecific control antibody anti-Histone H3.3 (Diagenode, Cat#: C15210011, final concentration 10 μg/ml) were added to 50 μl of beads resuspended in 1 ml of ACB overnight at 4 °C, 1 day prior to the RIP assay. To remove the uncoupled antibodies, beads were washed thrice with ACB and equilibrated in 100 μl of ACB.

Pre-cleared nuclear lysates were next incubated overnight at 4 °C with the beads coupled to the indicated antibody. After extensive washing with RIPA buffer, beads were subjected to Proteinase K treatment in the Reverse cross-linking buffer (100 mM NaCl, 10 mM Tris-HCl pH 7.0, 1 mM EDTA, 0.5% SDS, 1 mg/ml Proteinase K [Euromedex]) at 50 °C for 1 h with end to end shaking followed by an incubation at 95 °C for 10 min[44]. Trizol reagent (Invitrogen) was added to the bead mixture or input RNA for RNA isolation. Contaminating DNA was removed by the RNase-free Ambion™ DNAse I (Invitrogen) treatment. Reverse transcription was performed using Superscript III™ (Invitrogen) followed by quantitative RT-PCR using PowerUp™ SYBR® Green Master Mix.

The primers HDV-835-851 and HDV-905-889 used for HDV RT-qPCR are detailed in the Supplementary Table 1. Results are expressed as % of input. The "No Antibody" value was subtracted from all conditions.

For the identification of the HDV genomic or antigenomic strand in RIP samples, the RNAs isolated as described above were subjected to the Biotynilated Magnetic Beads-based qRT-PCR assay[20]. The RNA was denatured at 95 °C for 10 min with 0.5 μM of biotinylated HDV Forward and Reverse primers (see Supplementary Table 1) for genomic HDV RNA identification or antigenomic HDV RNA identification, respectively, and immediately cooled down to −20 °C. The ABI Fast 1-Step Virus Master mix (Applied Biosystems, Cat#: 4444436) was next added for the reverse transcription step and was performed at 50 °C for 5 min followed by enzyme inactivation at 95 °C for 20 s. Biotinylated cDNA purification was carried out using the MinElute PCR Purification Kit (Qiagen, Cat#: 28004) where the cDNA was eluted in a final volume of 20 μl of RNase-free water. Biotinylated cDNA isolation was carried out using the Dynabeads™ kilobase BINDER™ Kit (Invitrogen, Cat#: 60101) where 5 μl of Dynabeads were washed and resuspended in 20 μl Binding Buffer, incubated with 20 μl biotinylated cDNA for 3 h at room temperature on a thermal shaker, washed twice in Washing Buffer, and finally once with RNase-free water. Purified RNA was finally resuspended in RNase-free water. For qRT-PCR, 1 μl of purified biotinylated cDNA bound to Dynabeads with HDV-specific primers and probes and an ABI Fast Advanced Master (Applied Biosystems) were used under the following conditions: Initial step 95 °C 20 s; 40 cycles at 95 °C for 3 s and 60 °C for 30 s.

**Strep-Tag® S-HDAg purification and MS analysis**. HepaRG cells and HepaRG ST-S-HDAg cells were grown in T175 flasks until differentiation. To collect the nuclei of wt HepaRG and HepaRG ST-S-HDAg cells, collected cells were incubated in a hypotonic buffer (Hepes 10 mM pH 7.9, MgCl$_2$ 1.5 mM, EGTA 0.1 mM, DTT 0.5 mM, Glycerol 5%, and cOmplete™ Mini Protease Inhibitor Cocktail [Roche]) for

30 min and centrifuged at 2500 × *g* for 15 min. Nuclear pellets were washed in the same buffer, cleared by centrifugation at 2500 × *g* for 15 min at 4 °C and lysed using the M–PER (Thermo Scientific) reagent supplemented with Benzonase® nuclease (Sigma Aldrich, final concentration 1 U/ml) and MgCl$_2$ (1 mM). Strep-Tactin® charged magnetic agarose beads (Qiagen) were added to the nuclear protein extract and incubated at 4 °C for 2 h. Beads were washed three times with Tris-HCl 0.8 M pH 8.0, NaCl 1.2 M, and EDTA 8 mM, Tween-20, resuspended in Laemmli (2×) and incubated at 95 °C for 5 min. Eluted samples were separated on a 12% SDS poly-acrylamide gel and incubated in a staining solution containing 0.1% Coomasie Brilliant Blue R–250 dye (Thermo Fisher Scientific), 40% ethanol, 10% acetic acid at 60 °C for 1 h. Gels were destained in Ethanol (10%) and Acetic acid (7.5%) under gentle agitation and stored in deionized water at 4 °C until MS analysis. MS analysis was performed at the "Protéomique Structurale et Fonctionnelle" platform (Institut Jacques Monod University Paris 7, France). Briefly, each lane was cut in 5 spots. Proteins in spots were reduced with DTT 10 mM, alkylated with iodoacetamide 55 mM, digested overnight at 37 °C by trypsin (12.5 μg/ml; Promega) in 20 μl of NH$_4$HCO$_3$ 25 mM and extracted from the gel. Following an Easy nano Liquid Chromatography (LC 1000 system), digests were analyzed by a LTQ Velo-sOrbitrap (Thermo Fisher Scientific). Data were processed with the Proteome Discoverer 1.2 software (Thermo Fisher scientific) coupled to an in-house Mascot search server (Matrix Science, version 2.3.02). MS/MS data were searched against SwissProt databases with the *Homo sapiens* taxonomy and NCBInr with the viruses taxonomy.

**Pull-down assays**. Huh7 cells were transiently transfected with the plasmid pEXPR105-ST-S-HDAg or pEXPER105-S-HDAg. Cells were collected 3 days post transfection and pelleted by centrifugation at 1500 × *g* for 5 min. Cells pellets were incubated in NIB buffer supplemented with 20 mM Sodium Butyrate for 30 min under constant shaking on ice followed by dounce homogenization. After centrifugation at 2500 × *g* for 15 min at 4 °C, the supernatant was discarded and nuclear pellets were washed once with PBS supplemented with cOmplete™ Mini Protease Inhibitor Cocktail (Roche), PMSF (1 mM) and were cleared by centrifugation at 2500 × *g* for 15 min at 4 °C. Nuclear pellets were resuspended in RIPA buffer supplemented with Benzonase® nuclease (Sigma Aldrich, 1 U/ml), 20 mM Sodium Butyrate and 1 mM MgCl$_2$ for 15 min on ice. Nuclear lysates were then transferred to bioruptor microtubes (Diagenode) and sonicated (30 sec on and 45 sec off, 2 cycles) using a Diagenode Bioruptor® pico water bath sonicator. Nuclear lysates were next cleared by centrifugation at 13,000 × *g* for 10 min at 4 °C. A fraction of the cleared nuclear lysates was kept aside as "input". The remaining lysates were divided into fractions and incubated with 20 μl packed bead volume of Strep-Tactin® XT magnetic beads (IBA, Cat#: 2–4090–002) that had been washed 3 times in the Strep-Tactin® XT wash buffer (IBA, Cat#: 2–1003–100). Beads were incubated overnight with constant rotation at 4 °C, washed 3 times with 1 ml RIPA to remove non-specifically bound proteins and resuspended in 500 μl of the same buffer. For silver staining analysis (Pierce™ Silver Stain kit, Thermo Scientific) and immunoblotting, a fraction of beads was removed here and eluted in the Laemmli (2×) for 5 min at 95 °C.

In the pull-down assays shown in Fig. 2g, Strep-Tagged S-HDAg proteins immobilized on the Strep-Tactin® resin were incubated either with His6-BAZ2B BRD (Cat#: Abcam, Cat#: 196402) or His6-GFP (Abcam, Cat#: 134853) for 3 h at room temperature on a rotating wheel. Following extensive washes with the nuclei lysis buffer, bound proteins were released by incubation with Laemmli (2×) for 5 min at 95 °C. The eluates were resolved on 4–20% Tris-HCl Criterion™ Precast gels (Bio-Rad) to assess Strep-tagged protein expression, purification and His6-tagged protein binding by immunoblotting. In the pull-down experiments shown in Supplementary Fig. 4c, equimolar amounts of His6-BAZ2B BRD (4 μg) and His6-GFP (12 μg) proteins were immobilized on the Ni-NTA agarose beads (Qiagen, Cat#: 361111) and then incubated with purified ST-S-HDAg for 3 h at room temperature on a rotating wheel. Following extensive washes in a buffer containing 100 mM Tris-HCl, pH 8.0, 5% Glycerol, 0.5 mM DTT, and 5 mM imidazole, bound proteins were released by incubation with Laemmli (2×) for 5 min at 95 °C. The eluates were resolved on a 4–20% precast SDS polyacrylamide gel and analyzed by immunoblotting.

**Quantification and statistical analyses**. Unless otherwise stated, all experiments were performed at least three times and all data involving statistics are presented as mean ± SEM. A non-parametric two-tailed unpaired Kruskal–Wallis with a multiple comparison correction and analysis of variance tests were used for statistical analysis. GraphPad Prism version 7.05 for Windows (GraphPad Software, San Diego, CA) was used to create graphs.

## Data availability
The mass spectrometry proteomics data have been deposited to the ProteomeXchange Consortium via the PRIDE partner repository with the dataset identifier PXD016880 and to the IMEx consortium (www.imexconsortium.org) through IntAct[49], and assigned the identifier IM-27520. All other data, materials, and reagents are available on request from the corresponding author (M.L.). The source data underlying Fig. 1, Fig. 2, Figs. 3b–f, Fig. 4, Fig. S1, Fig. S4, and Fig. S5 are provided as a Source Data file.

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

## Acknowledgements

We thank Alan Campbell Kay for the gift of anti-delta antibodies and Michel Rivoire (Centre Léon Bérard, Lyon) for providing HBV, HCV, and HIV negative human liver resections. We thank Christophe Combet for bioinformatic support. We thank Sebastien Violot (IBCP, Lyon, France), Georges Abou-Jaoudé (INTS, Paris, France), Alan Campbell Kay, and Fabien Zoulim (CRCL, Lyon, France) for fruitful discussions and advice. We thank Sophie Clement for her help in confocal microscopy imaging. P.D. was supported by an INSERM Interface Contract (2007–2011). N.A.S. was the recipient of a PhD grant from the French Ministry of Research and Technology (2013–2016) and a fellowship from the ANR@RACTION (2017–2018). This work was supported by grants from the Agence Nationale pour la Recherche sur le SIDA et les hépatites virales (ANRS) to P.D. (AO 2011–1 and AO 2013–1) and to M.L. (numbers ECTZ8323; ECTZ27696; ECTZ66014), from the Agence Nationale de la Recherche (ANR@TRACTION) to M.L.; from the EU project 667273 HEP-CAR to M.L. S.M.K., A.P. and M.S. are supported by the SGC, a registered charity (number 1097737) that receives funds from AbbVie, Bayer Pharma AG, Boehringer Ingelheim, Canada Foundation for Innovation, Eshelman Institute for Innovation, Genome Canada, Innovative Medicines Initiative (EU/EFPIA) [ULTRA-DD grant no. 115766], Janssen, Merck KGaA Darmstadt Germany, MSD, Novartis Pharma AG, Ontario Ministry of Economic Development and Innovation, Pfizer, São Paulo Research Foundation-FAPESP, Takeda, and Wellcome [106169/ZZ14/Z].

## Author contributions

J.C.C., M.L., and P.D.: designed the study, directed its implementation, analyzed and interpreted the data, and finalized the manuscript. N.A.S.: performed experiments, analyzed the data, and drafted the manuscript. C.S.: contributed to the HDV infection experiment, analysis of results, and revision of the manuscript. D.A.: generated and characterized the HDAg, HepaRG cell lines, and contributed to the revision of the draft manuscript. S.M., A.C., and M.S.: contributed to BAZ2B functional experiments and reviewed the draft manuscript. F.G. contributed to the RIP assays and reviewed the draft manuscript. P.M. contributed to implementation of the study and reviewed the draft manuscript.

## Competing interests

The authors declare no competing interests.
