## [Peer Review File · Nature Communications]

Reviewers' comments:

Reviewer #1 (Remarks to the Author):

The manuscript from Natali Abeywickrama-Samarakoon addresses the mechanism of jHepatitis Delta virus (HDV)–mediated control of the host transcriptional machinery that supports viral replication. I find the manuscript exciting, novel and very clear. I would recommend the publication in its current form and will be happy to write commentary, if appropriate.

Reviewer #2 (Remarks to the Author):

Abeywickrama-Samarakoon et al present a well-executed study where they show that the S-HDAg protein from HDV interacts with BAZ2A in cells. They claim that the interaction is dependent on acetylation and that this represents a histone mimic by the virus to hijack the host protein. Although the data presented are clear and easy to follow, the conclusions are not warranted. The authors present in vitro binding ICT assays where they show that the acetylated peptide motif of the viral protein clearly does not bind to the bromodomain of BAZ2A. This result is in contradiction with their model where the viral S-HDAg binds BAZ2A as a histone mimic, which would require a direct interaction. The idea that viral proteins mimic histones and hijack and manipulate chromatin remodelers is an important one, with broad implications. However, without resolution of this crucial point, the claims of the paper are not supported.

Major concerns:

- I disagree with the use of the term 'pseudochromatin'. This suggests a DNA-protein complex that mimics chromatin in a different way. Griffin et al 2014 are careful to make an analogy when describing the ribonucleoprotein complex as being similar to nucleosomes, but the use of 'pseudochromatin' multiple times in this manuscript is misleading.

- I am not convinced that the interaction between S-HDAg and BAZ2A is direct. From the data presented here, the interactions are based in cells and immunoprecipitations with nucleases, but it is entirely possible that the interaction is dependent on the presence of other proteins and not direct.

- What is the evidence that BAZ2A can activate pol II?

- Sup Figure 1e lane 5 is a smear, one cannot conclude that the HDV AG RNA band is not there when there is such a smear, it could easily be masked. This should be repeated and further clarified to make the conclusions claimed by this result.

- What is the positive control for the mass spec? It is unclear whether the pull-down of S-HDAg with mass spec was also done in the context of infection, or if any known interactors were also pulled down. Also, some statistics or other validation of the mass spec would be helpful.

- Cell culture experiments are not *in vivo* and should not be stated as such. Throughout the paper, the authors refer to any cell experiments as '*in vivo*' but no animal model is used in this paper and whether the results observed would hold true in tissue or in an animal model cannot be assumed. This should be corrected throughout the text.

- Is the 4 amino acid motif conserved across other HDV serotypes?

- Figure 2 is entirely a negative result that does not support the hypothesis that acetylated S-HDAg is bound by BAZ2A. This suggests an indirect interaction in cells that is dependent on another mediator not present in the *in vitro* experiment – suggesting the model is wrong. *In vitro* pull down experiments with the full proteins (e.g. recombinant) may show an interaction that warrants the model. This is a key experiment to claim a histone mimic as BAZ2A clearly directly interacts with H3.

- It is apparent from Figure 3 that the region in S-HDAg is important for the interaction with BAZ2A-BRD in cells, but not that this is a direct interaction. Again the use of '*in vivo*' to describe cell culture experiments is incorrect and misleading, no experiments were done in any animal model or human tissue samples.

- In figure 4, the mutation in S-HDAg does impact viral RNA accumulation but it is a modest defect (at best 3 fold) that the virus recovers from by day 9. There are no statistics in panel A, is this statistically significant? A reproducible result? If the virus recovers over time, this also suggests other redundant mechanisms that overcome the mutation and suggests this finding may not be crucial to virus infection.

Minor concerns:

- page 1 line 3, 'able' should be 'capable'

- second sentence is extremely long and hard to follow

- How does DNA dependent RNA polymerase make RNA from an RNA template if it is DNA-dependent? In the introduction this contradiction is not addressed or explained. For a wide audience this should be clarified.

- The conclusion that BAZ2A-BRD is a druggable target of HDV is a provocative one, however, BAZ2B has many functions that are not explored here and targeting this protein could have detrimental off target effects.

Reviewer #3 (Remarks to the Author):

In their manuscript, Abeywickrama-Samarakoon et al. perform a mass spec based screen for interaction partners of the Hepatitis Delta antigen. They identify BAZ2B, which is a member of the BRF chromatin remodeling complex, as a binding partner and map the binding site to the acetylated Lys residue K72 motif within HDAg, which is similar to the authentic binding motif within the histone 3 tail. The authors further claim that abrogation of this binding leads to a decrease in viral replication. Overall, the novelty of this study is given, this possible interaction has never been described before. In general, the topic is interesting not only to HDV virologists but also to a more general readership, as HDV is the only virus that hijacks a cellular DNA-dependent RNA polymerase for its own viral RNA replication, a peculiar mechanism, which is still not understood, but the authors here give a first evidence to a possible mechanism of recruitment of the polymerase. The biochemical data of the manuscript is convincing, all binding experiments seem to be properly controlled, however, the virological data is very poor, relying on plasmid transfection rather than authentic infection experiments, although the authors seem to have all relevant tools already in hand (cell lines HepaRG, PHH, etc.). In order to justify publication in Nature Communications, the virological part needs to be improved and the role of the BAZ2B-HDAg interaction during authentic infection needs to be clarified.

Major points:

1. The authors need to show the relevance of their proposed interaction during HDV infection. The data in Fig. 4 based on plasmid transfection in HDAg-overexpressing cells is experimentally rather poor and so are the observed effects. Cell lines (HepaRG, HuH7(-NTCP?), PHH) seem to be available in the author's lab. Can you knockdown BAZ2B by siRNA/shRNA/small molecule(?) and does this lead to a decrease in viral replication after infection? Can you do Co-IP in infected cells to proof interaction between BAZ2B-HDAg (similar to Fig1d, but looking at protein-protein rather than protein-RNA interaction)? Can you, by confocal microscopy, show co-localisation between HDAg & BRF in HDV-infected cells? Does the localization pattern of BRF change in infected versus non-infected cells?

2. What exactly is the pSVLD2m plasmid and which mutation does it encode? On page 4, the authors state that pSVLD2m is a replication-defective plasmid that cannot produce S-HDAg and replication is only initiated when S-HDAg is provided in trans. In the methods section, it is also stated that the plasmid was provided by John Taylor. Unfortunately, the authors do not include a single citation for these statements. Literature research did not reveal any publications by John Taylor using this plasmid, however, a plasmid with the same name was described by Chang et al., PNAS, 1991. This particular plasmid has a frameshift mutation in the L-HDAg, but expresses S-HDAg and leads to viral replication but not assembly of virions. It is unclear to the reviewer, which plasmid was used in the present study. Also, the results obtained with this plasmid are unclear: Fig. S1e, lane 5 clearly shows viral RNAs after transfection of the plasmid, in the absence of HDAg. There might even be a band at 1.7kb, which is not visible due to signal saturation with the shorter products. Fig. 4a: here, an important control is missing: transfection of the plasmid in HuH7 cells without HDAg to show that there is indeed no replication in the absence of HDAg. Anyways, the experiments in Fig. 4ab must be repeated with NTCP-transfection of the three cell lines HuH7, HuH7-HDAgwt, HuH7-HDAgR75A and subsequent HDV infection. If you then still observe the mentioned effect on HDV RNA and L-HDAg expression, this would clearly strengthen your point.

Minor points:

1. L-HDAg suppresses viral replication and might therefore stop the recruitment of pol II. Have you ever tested, if BAZ2B also binds L-HDAg?

2. The BRF complex is made up by several subunits including ATPases, which you nicely describe in the manuscript. BAZ2B seems to bind HDAg at K72 and HDAg in-turn binds viral RNA. This is all very complex and complicated with many different protein names that most readers probably have never heard before. Here a graphical scheme of how you believe the binding and composition of the complexes looks like would greatly help the reader in understanding.

3. Fig. 1d: You show that, when immunoprecipitating BAZ2B, you find viral RNA, probably associated to HDAg. Have you tested or can you speculate if this interaction is preferably with genomic or also with antigenomic RNA?

4. Fig. 4a: You claim that viral replication is decreased in the R75A cells compared to wt because of less BAZ2B-recruitment. Could it also be possible that the R75A-HDAg binds less efficient to viral RNA

than wt-HDAg, therefore decreasing viral replication. Can you test direct binding of your proteins to viral RNA, e.g. by co-immunoprecipitation?

5. Fig. 4a: bar charts, y-axis: the "1" in "100" of your scale is missing

6. Page 14, line 11: delete "s" from "interacts"

Point to point response to reviewers' comments

Reviewer #1.

We have been delighted to learn that Reviewer 1 found our manuscript “exciting, novel and very clear”.

Reviewer #2.

General comment

Abeywickrama-Samarakoon et al. present a well-executed study where they show that the S-HDAg protein from HDV interacts with BAZ2(B) in cells. They claim that the interaction is dependent on acetylation and that this represents a histone mimic by the virus to hijack the host protein. Although the data presented are clear and easy to follow, the conclusions are not warranted. The authors present in vitro binding ICT assays where they show that the acetylated peptide motif of the viral protein clearly does not bind to the bromodomain of BAZ2(B). This result is in contradiction with their model where the viral S-HDAg binds BAZ2(B) as a histone mimic, which would require a direct interaction. The idea that viral proteins mimic histones and hijack and manipulate chromatin remodelers is an important one, with broad implications. However, without resolution of this crucial point, the claims of the paper are not supported.

We were happy to read that Reviewer 2 found our study ‘well executed and ‘the idea that viral proteins mimic histones and hijack and manipulate chromatin remodelers is an important one, with broad implications’. We also appreciated the challenging quality of the comments that prompted us to work more to further and better support our conclusions.

As detailed in the answers to the specific questions from both Reviewer 2 and Reviewer 3, we have performed a number of additional experiments to support the notion of a specific and direct interaction between the BAZ2B BRD and S-HDAg. In particular, we have performed *in vitro* experiments to pull-down a recombinant His₆-tagged-BAZ2B bromodomain (BRD) using a full-length StrepTag-S-HDAg expressed in Huh7 cells and affinity purified in the presence of an HDACi to preserve K72 S-HDAg acetylation.

On a minor note, the protein we identified as an interactant of S-HDAg is BAZ2B and not BAZ2A. Both BAZ2A, also known as TIP-5 α , and BAZ2B contain a bromodomain but their known functions are quite different. Several reports implicate BAZ2A as part of the NORC repressive complex that regulates Pol I-dependent transcription of the rRNA cluster, both in normal cells and in cancer. In our answers we just substituted BAZ2B for BAZ2A.

Major points

Question 1. *I disagree with the use of the term ‘pseudochromatin’. This suggests a DNA-protein complex that mimics chromatin in a different way. Griffin et al 2014 are careful to make an analogy when describing the ribonucleoprotein complex as being similar to nucleosomes, but the use of ‘pseudochromatin’ multiple times in this manuscript is misleading.*

Answer 1. We used the term *pseudo-chromatin* twice in the original MS: in the Introduction section (page 3, lines 24-29) and in the Discussion section (page 14, line 14-16). In the first case, after describing how the HDV RNP is likely organized in the nuclei of HDV infected hepatocytes according to the work of Casey and coworkers (*‘In the current model of HDV ribonucleoprotein (RNP) complex organization, 4 to 5 octamers of the HDAg proteins are wrapped by the viral RNA to form nucleosome-like structures’*), we incorporated the new information coming from our MS and in particular the recruitment onto the HDV RNP of host chromatin remodelers : *‘According to this model, the HDV RNP would be organized as a compact pseudo-chromatin requiring S-HDAg acetylation and the intervention of cellular chromatin remodeling factors to create a setting compatible with RNA Pol II recruitment and activation of transcription from the HDV RNA template’*.

We acknowledge that the term *pseudo-chromatin* may in the first instance lead to think of a 'DNA-protein complex that mimics chromatin'. We are also well aware of how cautious Casey and co-workers have been (Griffin 2014, ref 13 in the original manuscript) in making 'an analogy when describing the ribonucleoprotein complex as being similar to nucleosome'. They summarized and commented their Atomic Force Microscopy (AFM) analysis of the interaction between HDV RNA segments with HDAG by writing '... (HDV) RNA is condensed, perhaps wrapped, in a manner that is reminiscent of the way DNA is condensed in nucleosomes'. It is noteworthy that the AFM and SHAPE experiments described in Griffin paper are *in vitro* experiments whereas we provide evidence for an interaction between the HDV RNP and the host BRF chromatin remodeling complexes in HDV-infected human primary hepatocytes. The use of the term *pseudo-chromatin* intended to portray in a simple *pictorial* way the new concept supported by our results, notably the capability of acetylated HDAG to recruit the BRF1/5 chromatin remodeling complexes through the interaction with the BAZ2B bromodomain in order to set the stage for Pol II recruitment on the HDV RNP template.

HDAG has been reported to stimulate transcription elongation by displacing negative elongation factor A (NELF-A) from Pol II (Yamaguchi et al. 2001) and by accelerating forward translocation of Pol II at the cost of fidelity (Nedialkov et al. 2003; Yamaguchi 2007). Based upon these observations generated in *in vitro* systems, HDAG has been described as a *viral* transcription elongation factor and it has been proposed that these properties of HDAG may contribute to the unusual RNA-dependent RNA synthesis by Pol II. Our results, obtained in relevant cellular models of HDV infection, provide new mechanistic insights to understand the engagement of Pol II in RNA-dependent RNA synthesis from HDV templates and the role of S-HDAG in the process.

In response to the concerns of *Reviewer 2*, we decided to avoid the term *pseudo-chromatin* in the revised MS. The sentence in page 3 (lines 30-34 in the revised manuscript) now reads 'According to this model, the RNP would adopt a chromatin-like organization where the viral RNA replaces the cellular DNA as template for HDV RNA synthesis by Pol II. This process is likely to require S-HDAG acetylation and the intervention of cellular chromatin remodeling factors to create a setting compatible with RNA Pol II recruitment and activity'.

We have also reconsidered the use of the terms *pseudo-minichromosome* (used in the title of the original MS) and *pseudo-chromosome* (used once in the discussion in the original manuscript) to describe the HDV RNP. While the term *viral mini-chromosome* is widely accepted, on the basis of the recruitment of cellular histones and non-histone proteins and the visualization of classical 'beads-on-a-string' nucleosome-like structures, to describe the HBV cccDNA, at this stage there is not enough evidence to claim a chromosome-like organization for the HDV RNP. We concluded that if one avoids the term *pseudo-chromatin* the same should apply to the terms *pseudo-minichromosome* and *pseudo-chromosome*. We have modified the manuscript accordingly and we propose to change the title of the MS from 'Hepatitis Delta Virus histone mimicry drives HDV pseudo-minichromosome formation and viral RNA progeny synthesis' to 'Hepatitis Delta Virus histone mimicry drives the recruitment of cellular BRF chromatin remodelers for viral RNA replication'.

Question 2. *I am not convinced that the interaction between S-HDAG and BAZ2(B) is direct. From the data presented here, the interactions are based in cells and immunoprecipitations with nucleases, but it is entirely possible that the interaction is dependent on the presence of other proteins and not direct.*

Answer 2. We acknowledge that this is a critical point and we agreed with *Reviewer 2* and the Editor to perform new experiments aimed to answer this question. See also Question 8 from *Reviewer 2* and Question 1 from *Reviewer 3*.

Altogether, in the revised manuscript we now present evidence from 3 different approaches to support a direct interaction between the S-HDAG KacRXR short linear interacting motif (SLiM) and the BAZ2B BRD :

i) we have performed *in vitro* experiments to pull-down the recombinant His₆-tagged-BAZ2B bromodomain (BRD) using as bait a full-length StrepTag-S-HDAG expressed in Huh7 cells and affinity purified in the presence of an HDACi to preserve K72 S-HDAG acetylation. A simple pull-down experiment performed with recombinant BAZ2B BRD and recombinant S-HDAG would not ensure a correct evaluation of the interaction between S-HDAG and BAZ2B because BRDs require an acetylated substrate for binding and recombinant S-HDAG would not be acetylated. The *in vitro* acetylation of recombinant HDAG could have been an alternative to the cumbersome purification protocol we adopted but S-HDAG would have needed to be purified anyway to eliminate the acetyltransferase used for the *in vitro* acetylation reaction.

An additional option could have been represented by the use of a recombinant S-HDAg carrying a K/Q substitution in position 72 in order to mimic the relevant lysine acetylation. However, BRDs need the presence of an acetylated residue to interact and do not efficiently bind *in vitro* to charge based acetylation mimics (Xu L et al 2017 and *unpublished observations*). Although we cannot formally exclude that our affinity purified StrepTag-S-HDAg expressed in Huh7 cells might contain traces of host cofactors that could participate to S-HDAg/BAZ2B interaction, the new data shown in Fig 2f of the revised manuscript (additional information in Supplementary Figure 3) support the notion of a specific and direct interaction between the BAZ2B BRD and S-HDAg and the role of S-HDAg histone mimicry in the HDV viral cycle. These results are described and commented in the new Results section '*BAZ2B acts as a host co-activator of HDV replication in PHHs*' (page 7 of the revised manuscript).

ii) the BAZ2B BRD inhibitor GSK-2801 greatly reduces the recruitment of components of the BRF1/5 remodeling complexes and of RNA Pol II on the HDV RNP. These new RIP experiments performed in HDV infected PHHs are shown in the Fig 2e of the revised manuscript and the results are described and commented in the new Results section '*BAZ2B acts as a host co-activator of HDV replication in PHHs*' (page 7 of the revised manuscript). The co-crystal structure of GSK-2801 and BAZ2B BRD has been resolved (Chen P, 2016 ; ref. 21 in the revised manuscript) and GSK-2801 has been shown to directly bind to the acetyl-lysine binding pocket of the BAZ2B BRD and acetylated K14 in histone H3. Based upon these observations, the RIP results shown in the new Fig 2e, while not formally excluding the contribution of additional proteins, further support a direct interaction between S-HDAg SLiM and BAZ2B BRD.

iii) the co-immunoprecipitation of GFP-BAZ2B-BRD and wild type S-HDAg is almost completely abrogated when the R75A HDAg is substituted to wild type S-HDAg (Fig 3d in the original manuscript; new Fig 3e). These results provide an additional circumstantial evidence in favor of a direct interaction between S-HDAg and BAZ2B BRD mediated by the S-HDAg KacRXXR motif.

Question 3. *What is the evidence that BAZ2(B) can activate pol II?*

Answer 3. We do not claim in any part of the submitted manuscript that BAZ2B "activates" Pol II. In page 2 (line 28) we wrote that the interaction between acetylated S-HDAg and chromatin remodeling complexes '*create(s) a setting compatible with RNA Pol II recruitment and activation of transcription on the HDV RNA template*'. In page 7 (line 9), while commenting on the RNA immunoprecipitation results, we wrote that '*the BRF host chromatin remodelers are associated with S-HDAg on Pol II-associated transcriptionally active, replicating HDV ribonucleoprotein (RNP)*'. Thus, we were cautious not to suggest a direct and causative link between the recruitment of the BRF complex and the recruitment of Pol II or, an even more far-fetched concept, that BAZ2B might directly activate Pol II enzymatic activity. On the other hand, we believe that, according to the consolidated evidence in the HDV literature, we can infer that Pol II, once recruited, is directing RNA synthesis from the HDV RNP template (i.e. HDV transcription and replication). The new RNA immunoprecipitation experiments showing a sharp reduction of Pol II recruitment onto the HDV RNP in HDV-infected PHHs treated with the BAZ2B bromodomain inhibitor GSK2801 (new Fig 2e) provide additional evidence that links the interaction between acetylated S-HDAg and BAZ2B to Pol II recruitment, and hence to viral RNAs synthesis from the HDV RNP, leading to increased viral transcription and replication.

As already mentioned above (answer to *Question 1*) HDAg has been reported to stimulate transcription elongation by Pol II in *in vitro* systems (Yamaguchi et al. 2001; Nedialkov et al. 2003; Yamaguchi 2007). Our results, obtained in relevant cellular models of HDV infection, provide new insights to understand the engagement of Pol II in RNA-dependent RNA synthesis from HDV templates and the role of S-HDAg in the process. Indeed, the abrogation of BAZ2B expression by lentivirus-mediated transduction of specific shRNAs (new Fig 2a and 2b) and the inhibition of BAZ2B BRD activity by the small molecule inhibitor GSK2801 (new Fig 2c) lead to a strong reduction of HDV RNA synthesis. These results, together with the reduction of Pol II recruitment onto the HDV RNP in HDV-infected PHHs treated with GSK2801 (that inhibits the interaction between the BAZ2B BRD and its acetylated targets on H3), further support the notion that acetylated S-HDAg interaction with BAZ2B BRD favors Pol II recruitment and, as a consequence, Pol II driven HDV RNAs synthesis from the HDV RNP, but do not indicate that BAZ2B directly activates Pol II enzymatic activity.

Question 4. *Supplementary Figure 1e, lane 5 is a smear, one cannot conclude that the HDV AG RNA band is not there when there is such a smear, it could easily be masked. This should be repeated and further clarified to make the conclusions claimed by this result.*

Answer 4. We repeated this experiment as requested by the Reviewer. To confirm that the ST-S-HDAg protein is functional in HDV replication, Huh-7 cells were co-transfected with a StrepTag S-HDAg expression vector and the pSVL-D2M plasmid. As detailed in the Answer to Question 2 from Reviewer 3, pSVL-D2M allows the transcription of a full length HDV RNA but doesn't code for a functional S-HDAg and needs to be transcomplemented by S-HDAg. The new Supplementary Figure S1e shows no HDV RNA replication in cells are transfected with pSVL-D2M alone (lane 2), but it is restored when co-transfected with StrepTag S-HDAg expression vector (lane 1). Similarly, doxycycline induction of ST-S-HDAg expression in the tetracycline inducible ST-S-HDAg HepaRG cell line trans-complemented for the replication defective pSVL-D2M plasmid (lane 4). The replication competent pSVLD3 plasmid was used as positive control (lane 3).

Question 5. *What is the positive control for the mass spec? It is unclear whether the pull-down of S-HDAg with mass spec was also done in the context of infection, or if any known interactors were also pulled down. Also, some statistics or other validation of the mass spec would be helpful.*

Answer 5. The initial pull-down experiments were not performed in the context of HDV infection but in a lentiviral-transduced HepaRG stable cell line expressing a recombinant Strep-Tag S-HDAg protein (ST-S-HDAg) in a doxycycline inducible manner. Differentiated HepaRG cells are permissive for HDV infection and resemble differentiated human hepatocytes. The results of the mass spectrometry experiments are now described in more detail in the Results section 'Affinity capture/MS identifies BAZ2B as an interactant of the S-HDAg protein' (page 5 of the revised manuscript). The screening identified 270 proteins with a Mascot score > 20 and confirmed 15 proteins (listed in the Supplementary Table 1 of the revised manuscript) previously reported to interact with S-HDAg in HEK293 cells expressing a Flag-Tag-S-HDAg bait (Cao et al 2009). BAZ2B appeared among the proteins with the highest Mascot score co-purifying with S-HDAg. The peptide mass fingerprinting analysis identified 24 unique peptides spanning across the full length of the BAZ2B protein (see the Supplementary Fig. 2a in the revised manuscript) and 6 and 11 unique tryptic peptides spanning the full-length SNF2L (P28370) and SNF2H (O60264) ATPases (new Supplementary Figure 3). SNF2L and SNF2H are part of the BRF-1 and BRF-5 chromatin remodeling complexes (Oppikofer et al. 2017; ref 18 in the revised manuscript). Four additional peptides were present in both SNF2L and SNF2H proteins (new Supplementary Figure 2b).

Question 6. *Cell culture experiments are not in vivo and should not be stated as such. Throughout the paper, the authors refer to any cell experiments as 'in vivo' but no animal model is used in this paper and whether the results observed would hold true in tissue or in an animal model cannot be assumed. This should be corrected throughout the text.*

Answer 6. We substituted throughout the MS 'in vivo' with 'in cell culture' or similar wording when appropriate.

Question 7. *Is the 4 amino acid motif conserved across other HDV serotypes?*

Answer 7. Indeed, K72 and R75 are conserved in all the 8 HDV genotypes (see references 22 and 23 in the original manuscript or 26 and 27 in the revised manuscript). HDV strains are classified among eight genotypes (Le Gal et al., 2006). This information was included in the original manuscript (page 7, lines 23 to 27): 'The alignment of 274 S-HDAg sequences showed a perfectly conserved SLiM motif among all HDV genotypes. The K72 acetylation occurs precisely in the sequence motif K72ac-R/K-X-R75 (where X: is A, P, S or L)' and in Fig. 2a. In the revised MS the sentence (page 8, lines 2 to 4) was re-worded as follows: 'The alignment of 274 S-HDAg sequences showed a perfectly conserved SLiM motif across all 8 genotypes with both K72 (K73 in HDV-3) and R75 (R76 in HDV-3) residues conserved in all isolates from the eight HDV clades'.

Question 8. *Figure 2 is entirely a negative result that does not support the hypothesis that acetylated S-HDAg is bound by BAZ2(B). This suggests an indirect interaction in cells that is dependent on another mediator not present in the in vitro experiment – suggesting the model is wrong. In vitro pull-down experiments with the full proteins (e.g.*

recombinant) may show an interaction that warrants the model. This is a key experiment to claim a histone mimic as BAZ2(B) clearly directly interacts with H3.

Answer 8. As already stated in the answer to Question 2 we have performed the *in vitro* pull-down experiments suggested by the *Reviewer 2*. The results are described and commented in the new Results section 'BAZ2B acts as a host co-activator of HDV replication in PHHs' (page 7 of the revised manuscript). We used as bait a full-length StrepTag-S-HDAg expressed in Huh7 cells and affinity purified in the presence of an HDACi to preserve K72 S-HDAg acetylation (required for BRDs interaction) to pull-down the recombinant His₆-tagged-BAZ2B BRD (new Fig 2e).

These results, together with :

a) the reduction of the recruitment onto the HDV RNP of BRF1/5 components in HDV-infected PHHs treated with the BAZ2B BRD inhibitor GSK-2801 (see the answer to Question 2 from *Reviewer 2* for more details) and

b) the almost complete abrogation of GFP-BAZ2B-BRD and the R75A S-HDAg mutant co-immunoprecipitation (Fig 3d in the original manuscript; new Fig 3e), support a direct interaction between the S-HDAg KacRXR short linear interacting motif (SLiM) and the BAZ2B BRD.

On the other hand, we agree with the *Reviewer 2* that the Isothermal Titration Calorimetry (ITC) experiments (Fig 2b-g in the original manuscript), while showing the binding of BAZ2B-BRD to the SNF2H/L peptide KTIGYKacVPRNP (K_d=29.6 μM) and the H3 peptide KSTGGKacAPRKQ (K_d=6.0 μM), did not detect an interaction between BAZ2B-BRD and the S-HDAg peptide GAPPKacRARTD and can, therefore, be considered '*an entirely a negative result*'.

This affirmation is partially mitigated by:

a) the results obtained using 3 additional peptides in which the S-HDAg motifs KacKPR and KacRPR and the SNF2L/H motif KacVPR substituted the KacAPR in the H3 peptide backbone showing binding affinities and thermodynamic signatures indicating a physical interaction for all the H3/SHDAg SLiM and the H3/SNF2L/H SLiM hybrid peptides (Fig 2d-f in the original manuscript). Notably, the binding affinity value of the H3/SNF2L/H SLiM hybrid peptide (Fig 2g in the original manuscript) was higher than that of the wild type SNF2H/L peptide (Fig 2d in the original manuscript) (K_d=20.4 μM vs K_d=29.6 μM) and that the H3/SHDAg SLiM hybrid peptide (Fig 2f in the original manuscript) binds to the BAZ2B BRD and displays the highest binding affinity value (K_d=11.5 μM) after the wild type H3 peptide (Fig 2b in the original manuscript) (K_d=6.0 μM).

b) the appreciation of ITC limits in terms of the length of the peptides that can be used to perform the assays, with the consequent possible lack of important 3D conformational configurations, or the need for allosteric changes in the 3D conformation induced either by additional post-translational modifications (PTMs) or by the contact with other proteins and/or nucleic acids. By comparing the sequence of the different peptides and performing *in silico* modeling of the impact of the AA sequence around the SLiM on the interaction with the BAZ2B BRD it became evident that (Martin Schroeder, unpublished observations):

a) changes in one or both G residues in position -1/-2 before the SLiM reduces the binding affinity ;

b) the presence of the 2 P residues in position -2/-3 in the S-HDAg sequence imposes a strong spatial constrain on the interaction with the BAZ2B BRD;

c) changes in position +1 are more easily accomodated even if this penalizes to a variable extent the binding affinity (H3 K14(Ac)APR, K_d=6.0 μM; H3 K14(Ac) A15R, K_d=18.0 μM; H3 K14(Ac) A15V K_d=20.0 μM; H3 K14(Ac) A15K K_d=11.0 μM).

Whereas adding additional residues at the N-terminus will not likely result in a more potent binding affinity in the ITC assay, it is likely or at least highly possible that the conformation adopted by the partially disordered full-length S-HDAg allows, with or without the contribution of allosteric changes imposed by PTMs, the interaction with the BAZ2B BRD.

However, due to the controversial interpretation of the ITC data, the already mentioned limitations of the ITC assays and the availability now of additional experiments that support the specific interaction between the S-HDAg SLiM and the BAZ2B BRD (i.e. the mode of action of GSK-2801 and the results of the RNA immunoprecipitation in the presence of GSK-2801; the *in vitro* pull down experiments) we decided to remove the ITC data in the revised version of the manuscript.

Question 9a. *It is apparent from Figure 3 that the region in S-HDAg is important for the interaction with BAZ2A-BRD in cells, but not that this is a direct interaction.*

Answer 9a. This is an important point and, as already mentioned in our answer to Question 2, we performed two new sets of experiments to answer this question (see also Question 1 from Reviewer 3). First, we have performed *in vitro* pull-down experiments using a recombinant His₆-tagged-BAZ2B BRD as bait for affinity purified full-length StrepTag-S-HDAg expressed in Huh7 cells (new Fig 2f ; results described and commented in the new Results section '*BAZ2B acts as a host co-activator of HDV replication in PHHs*' at page 7 of the revised manuscript). Second, we have performed new RIP experiments in HDV infected PHHs showing that the BAZ2B BRD inhibitor GSK-2801, that interferes with the direct interaction between BAZ2B BRD and acetylated K14 in histone H3, greatly reduces the recruitment of components of the BRF1/5 remodeling complex and of RNA Pol II on the HDV RNP (new Fig 2e; results described and commented in the Results section '*BAZ2B acts as a host co-activator of HDV replication in PHHs*' at page 7 of the revised manuscript). These results, together with the co-immunoprecipitation experiments shown in new FIG 3e (Fig 3d in the original manuscript) support a direct interaction between S-HDAg and BAZ2B BRD mediated by the S-HDAg KacRXR motif.

Question 9b. *Again the use of 'in vivo' to describe cell culture experiments is incorrect and misleading, no experiments were done in any animal model or human tissue samples.*

Answer 9b. As stated in the answer to question 6, we substituted throughout the MS '*in vivo*' with '*in cell culture*' or similar wording when appropriate.

Question 10. *In figure 4, the mutation in S-HDAg does impact viral RNA accumulation but it is a modest defect (at best 3 fold) that the virus recovers from by day 9. There are no statistics in panel A, is this statistically significant? A reproducible result? If the virus recovers over time, this also suggests other redundant mechanisms that overcome the mutation and suggests this finding may not be crucial to virus infection.*

Answer 10. The results were reproducible and statistically significant. Fig 4a now displays a more informative Northern blot image and the histograms in the lower panels of new Fig 4a incorporate p values for both genomic and antigenomic HDV RNAs. To further probe the relevance of S-HDAg interaction with BAZ2B BRD and the effects of the R75A mutation in relevant cellular models of HDV replication (see also Question 1 from Reviewer 3), we have performed infection experiments in Huh7-106 cells (expressing the NTCP receptor) and in PHHs. To this aim we have produced HDV virions containing a genome coding for the R75A mutant S-HDAg (that lacks the capacity to interact and to be co-precipitated with the BAZ2B BRD, new Fig 3e). The infection experiments conducted in the hNTCP-expressing Huh7-106 cell line (HDV-susceptible) demonstrate a five-fold reduction of R75A HDV RNA replication as compared to wt (new Supplementary Figure 5c), whereas in primary cultures of human hepatocytes the effect is more pronounced with a near 2 log reduction for R75A HDV compared to wild type (new Fig 4c).

Minor points

Question 11. *page 1 line 3, 'able' should be 'capable'*

Question 12. *second sentence is extremely long and hard to follow*

Question 13. *How does DNA dependent RNA polymerase make RNA from an RNA template if it is DNA-dependent? In the introduction this contradiction is not addressed or explained. For a wide audience this should be clarified.*

Answer 11/12/13. The first paragraph of the introduction section has been extensively revised in order to address the specific concerns raised by the Reviewer 2, to be both more reader friendly and to provide, at the same time, all the relevant information for a general audience to understand the peculiar viral cycle of HDV.

Question 14. *The conclusion that BAZ2(B)-BRD is a druggable target of HDV is a provocative one, however, BAZ2B has many functions that are not explored here and targeting this protein could have detrimental off target effects.*

Answer 14. We agree that the functions of BAZ2B and the BRF1/5 complexes are under-explored and, therefore, targeting the BAZ2B-BRD may lead to unexpected "*detrimental off target effects*". This is a common phenomenon in all therapies targeting cellular functions and, in particular, when "*host targeting agents*" are tested as anti-viral agents whereas direct antivirals targeting virus specific functions needed

for viral replication should, at least in theory, be deprived of unwarranted effects on the host. Direct cellular toxicity and the impact on mitochondrial functions represent an early “no-go” signal in drug development. The full evaluation of the toxicity profile as well as the evaluation of the therapeutic margin of a new agent is part of its late pre-clinical and early clinical evaluation. Reviewer 2 is right when he reminds us that we are not at that stage. Indeed, we are providing new insights into the HDV viral cycle that identify potential therapeutic targets. This is particularly important in the case of HDV that is considered an ‘orphan disease’ from the therapeutic point of view and the European Medicines Agency and the US Food and Drug Administration granted new compounds to treat hepatitis D an orphan drug status. As we state in the ‘Discussion’ section ‘*Pegylated-interferon, the only authorized treatment for HDV-HBV co-infected chronic patients, may control the disease but has limited long-term effects the unique HDV life cycle and its dependence on HBV for its entry and egress are exploited for antiviral intervention. The HBV entry inhibitor Myrcludex B®, the HBsAg secretion inhibitors REP2139® and REP2165®, and the viral assembly and release inhibitor Lonafarnib, which are in early clinical development, all interfere with HBV envelope functions or the interaction between HDAg and the HBV envelope. Direct inhibition of HDV RNA replication has not been achieved so far*’. In this context, we think that pointing out that we have identified a potentially druggable step in the HDV viral cycle is important to put in the right context the relevance of our work. However, the warning regarding the potential for off targets effect stands and this led us to mitigate the last sentence of the Discussion section as follows: ‘*The recruitment of BRF complexes onto the HDV RNP and the role of S-HDAg interaction with the BAZ2B-BRD in HDV replication underlines the involvement of additional cellular factors, besides the Pol II and its partners in the host basal transcriptional machinery, and may help to identify a new potentially druggable targets for HDV*’.

Reviewer 3.

In their manuscript, Abeywickrama-Samarakoon et al. perform a mass spec based screen for interaction partners of the Hepatitis Delta antigen. They identify BAZ2B, which is a member of the BRF chromatin remodeling complex, as a binding partner and map the binding site to the acetylated Lys residue K72 motif within HDAg, which is similar to the authentic binding motif within the histone 3 tail. The authors further claim that abrogation of this binding leads to a decrease in viral replication. Overall, the novelty of this study is given, this possible interaction has never been described before. In general, the topic is interesting not only to HDV virologists but also to a more general readership, as HDV is the only virus that hijacks a cellular DNA-dependent RNA polymerase for its own viral RNA replication, a peculiar mechanism, which is still not understood, but the authors here give a first evidence to a possible mechanism of recruitment of the polymerase. The biochemical data of the manuscript is convincing, all binding experiments seem to be properly controlled, however, the virological data is very poor, relying on plasmid transfection rather than authentic infection experiments, although the authors seem to have all relevant tools already in hand (cell lines HepaRG, PHH, etc.). In order to justify publication in Nature Communications, the virological part needs to be improved and the role of the BAZ2B-HDAg interaction during authentic infection needs to be clarified.

We have been very happy that this Reviewer considered our MS novel and of potential interest ‘*not only for HDV virologists but also to a more general readership, as HDV is the only virus that hijacks a cellular DNA-dependent RNA polymerase for its own viral RNA replication, a peculiar mechanism, which is still not understood, but the authors here give a first evidence to a possible mechanism of recruitment of the polymerase*’. He also felt that the ‘virological data’ supporting the role of S-HDAg interaction with the BAZ2B BRD and the recruitment of the BRF1/5 remodeler complexes on the HDV RNP for HDV replication were rather ‘poor’ and he prompted us to expand our observations to relevant cellular models of HDV infection. We gladly accepted these criticisms and performed new experiments in the context of HDV infection to confirm the relevance of S-HDAg interaction with the BAZ2B BRD during HDV infection that are detailed in the answer to Question 1.

Major points

Question 1. *The authors need to show the relevance of their proposed interaction during HDV infection. The data in Fig. 4 based on plasmid transfection in HDAg-overexpressing cells is experimentally rather poor and so are the observed effects. Cell lines (HepaRG, HuH7(NTCP?), PHH) seem to be available in the author’s lab. Can you knockdown BAZ2B by siRNA/shRNA/small molecule(?) and does this lead to a decrease in viral replication after infection? Can you do Co-IP in infected cells to proof interaction between BAZ2B-HDAg (similar to Fig1d, but looking at protein-protein rather than protein-RNA interaction)? Can you, by confocal microscopy, show co-localisation*

between HDAG & BRF in HDV-infected cells? Does the localization pattern of BRF change in infected versus non-infected cells?

Answer 1. As mentioned above, we performed three new sets of experiments in the context of HDV infection in order to:

- assess the role of the SLiM motif in S-HDAg interaction with the BAZ2B BRD and the impact of the R75A mutation in the S-HDAg protein, that abrogates this interaction, on HDV replication. To this aim we produced R75A HDV virions (new Supplementary Figure 5a-b) and conducted HDV infections in the NTCP-expressing Huh7-106 cell line (new Supplementary Figure 5c) and in primary cultures of human hepatocytes (new Figure 4b and Supplementary Figure 5d). These new results show a five-fold reduction of R75A HDV RNA replication as compared to HDV wild type in Huh-106 cells whereas in PHHs, the the most relevant cellular model of HDV infection, R75A HDV RNA replication is reduced by a near 2-log factor in comparison to wild type HDV (new Fig 4b).

- evaluate the ability of the BAZ2B BRD inhibitor GSK-2801, that interferes with the direct interaction between BAZ2B BRD and acetylated K14 in histone H3 (Chen P, 2016 ; ref. 21 in the revised MS) and the recruitment of BAZ2B and the BRF1/5 complexes on the HDV RNP (new Fig. 2e), to inhibit HDV replication. In the new Fig 2c we show that GSK2801 treatment [10 µM] results in a significant reduction of HDV replication ($p < 0.0001$), in the absence of any significant cytotoxicity (new Fig. 2d). The control compound GSK8573, which has no effect on BAZ2B BRD (see ref 21 in the revised manuscript), did not affect HDV replication (new Fig. 2c).

- evaluate the contribution of BAZ2B to HBV replication by knocking down BAZ2B expression using shRNAs. As shown in the new Fig. 2, a 40 to 50% reduction of BAZ2B mRNA levels (new Fig. 2a) translated into a >50% inhibition of HDV replication at day 8 post-infection (new Fig. 2b).

Altogether, these results confirm the relevance of S-HDAg interaction with the BAZ2B BRD in the setting of HDV infection and the crucial role of the S-HDAg SLiM-like motif.

Finally, regarding HDAG and BRF co-localisation in HDV-infected cells we have been unable to generate this information because the BAZ2B antibody does not work in immunofluorescence or immunohistochemistry.

Question 2a. *What exactly is the pSVLD2m plasmid and which mutation does it encode? On page 4, the authors state that pSVLD2m is a replication-defective plasmid that cannot produce S-HDAg and replication is only initiated when S-HDAg is provided in trans. In the methods section, it is also stated that the plasmid was provided by John Taylor. Unfortunately, the authors do not include a single citation for these statements. Literature research did not reveal any publications by John Taylor using this plasmid, however, a plasmid with the same name was described by Chang et al., PNAS, 1991. This particular plasmid has a frameshift mutation in the L-HDAg, but expresses S-HDAg and leads to viral replication but not assembly of virions. It is unclear to the reviewer, which plasmid was used in the present study.*

Answer 2a. The pSVL-D2M plasmid is from John Taylor's laboratory and it is described in Kuo et al. J Virol. 1989, 63(5):1945-50 (ref 4 both in the original and in the revised manuscript) but it is not identified with a specific denomination in that paper. We now make reference to Kuo paper in the appropriate *Results* (page 5, line 14) and *Materials and Methods* (page 13, line 29) subsections. The name we use to identify this plasmid (pSVL-D2M) is the one provided to us by Mei Chao (a co-author of the above mentioned publication) on behalf of John Taylor at the time we signed the MTA to obtain it. The pSVL-D2M plasmid contains a dimer of the full-length HDV cDNA that is replication defective because it lacks S-HDAg coding capacity. Replication is rescued when co-transfected with a plasmid expressing a functional S-HDAg protein. We confirm that we did not performed any experiments with the plasmid pSVLDm2 (pSVL mutant number 2) described by Chang et al. (1991) that, as stated by *Reviewer 3*, cannot produce the L-HDAg protein and is defective for HDV assembly.

Question 2b. *Also, the results obtained with this plasmid are unclear: Fig. S1e, lane 5 clearly shows viral RNAs after transfection of the plasmid, in the absence of HDAG. There might even be a band at 1.7kb, which is not visible due to signal saturation with the shorter products.*

Answer 2b. As already mentioned in our answer to Question 4 from *Reviewer 2*, we repeated the experiment. The new Supplementary Figure S1e shows that HDV RNA replication is not observed when cells are transfected with pSVL-D2M alone (lane 2), but it is restored when co-transfected with StrepTag S-HDAg expression vector (lane 1), thus confirming the ability of ST-S-HDAg to trans-complement the

replication-defective pSVL-D2M. Similarly, doxycycline induction of ST-S-HDAg expression in the tetracycline inducible ST-S-HDAg HepaRG cell line trans-complemented for the replication defective pSVL-D2M plasmid (lane 4). In lane 3, the replication competent pSVLD3 plasmid is used as positive control.

Question 2c. *Fig. 4a: here, an important control is missing: transfection of the plasmid in HuH7 cells without HDAg to show that there is indeed no replication in the absence of HDAg. Anyways, the experiments in Fig. 4ab must be repeated with NTCP-transfection of the three cell lines HuH7, HuH7-HDAgwt, HuH7-HDAgR75A and subsequent HDV infection. If you then still observe the mentioned effect on HDV RNA and L-HDAg expression, this would clearly strengthen your point.*

Answer 2c. To answer to these concerns (see also our answers to the *General comment* and *Question 1* from *Reviewer 3*) we have produced R75A HDV virions (new Supplementary Figure 5a-b) and conducted HDV infection in the NTCP-expressing Huh7-106 cell line (new Supplementary Figure 5c) and in PHHs (new Figure 4b and Supplementary Figure 5d). We trust that the new results showing in PHHs, the most relevant cellular model of HDV infection, a near 2-log reduction of R75A HDV RNA replication as compared to wild type HDV (Fig 4b in the revised manuscript) '*strengthen our point*', as requested by the Reviewer.

Minor Points

Question 3. *L-HDAg suppresses viral replication and might therefore stop the recruitment of pol II. Have you ever tested, if BAZ2B also binds L-HDAg?*

Answer 3. No, this specific point has not been tested. Although interesting to investigate, the interpretation of binding - if any - would be complex since not only L-HDAg, contains the entire S-HDAg polypeptide but it also interacts and co-localizes with S-HDAg.

We focused on S-HDAg because it is involved in HDV RNP formation already in the early phases post-infection whereas L-HDAg, when its production starts, is more involved in HDV assembly rather than in Pol II recruitment. The translation of L-HDAg occurs after an RNA editing event on the HDV AG strand that changes the amber stop codon of S-HDAg to a tryptophan codon resulting in the addition of 19 or 20 additional aminoacids at the C-terminus. In the original manuscript (Fig. 4b) we showed that L-HDAg protein expression was delayed and reduced in Huh7 cells stably expressing S-HDAg R75A as compared to Huh7 cells stably expressing the wild type S-HDAg and we used this result to support the role of BRFs recruitment on the HDV RNP for HDV replication. The new results obtained in HDV infection models with the R75A HDV virus (new Fig. 4b and Supplementary Figure 2c-d), the anti-BAZ2B shRNAs (new Figure 2b) and with the BAZ2B BRD inhibitor GSK-2801 (new Fig 2c and 2e) led us to withdraw the old Figure 4b and the corresponding description in Results section.

Question 4. *The BRF complex is made up by several subunits including ATPases, which you nicely describe in the manuscript. BAZ2B seems to bind HDAg at K72 and HDAg in-turn binds viral RNA. This is all very complex and complicated with many different protein names that most readers probably have never heard before. Here a graphical scheme of how you believe the binding and composition of the complexes looks like would greatly help the reader in understanding.*

Answer 4. We have added a graphical scheme of the BRF1 and BRF5 complexes as Supplementary Figure 3 and we have added a second graphical scheme of how we model the interaction between S-HDAg and the BAZ2B BRD as Figure 5.

Question 5. *Fig. 1d: You show that, when immunoprecipitating BAZ2B, you find viral RNA, probably associated to HDAg. Have you tested or can you speculate if this interaction is preferably with genomic or also with antigenomic RNA?*

Answer 5. We performed strand-specific RT-PCR assay as described by Giersch et al. (ref 20 in the revised manuscript) coupled with the RNA immunoprecipitation assay and found that S-HDAg and P-Ser5 Pol II are recruited on both genomic and antigenomic HDV RNPs with similar efficiency whereas BAZ2B and SNF2H displayed a preferential binding on the HDV genomic strand (new Fig 1e-f). These results confirm the role of Pol II recruitment on the HDV RNP in both HDV mRNA synthesis from the genomic HDV RNA and the subsequent genomic RNA amplification from the antigenomic RNA template. The

preferential binding of BRF complexes on the HDV RNP containing genomic HDV RNA would suggest a role primarily, although not exclusively, in HDV '*transcription*'. In this context it is important to remind that, according to current knowledge, antigenomic HDV RNA synthesis and amplification from the genomic template is mediated by Pol I or Pol III and to occur in nucleoli. We could not confirm these results by Northern blot because the yield of HDV RNA in the RNA immunoprecipitation experiments is well below the sensitivity threshold of the technique. These results are described in the Results section '*BRF-1 and BRF-5 interact with the HDV RNP*' (page 6 line 34 and page 7 line 1-2).

Question 6. Fig. 4a: You claim that viral replication is decreased in the R75A cells compared to wt because of less BAZ2B-recruitment. Could it also be possible that the R75A-HDAg binds less efficient to viral RNA than wt-HDAg, therefore decreasing viral replication. Can you test direct binding of your proteins to viral RNA, e.g. by co-immunoprecipitation?

Answer 6. As shown in the new Supplementary Figure 5b, comparable amounts of wild type R75A and wild-type S-HDAg proteins can be found in wild type HDV RNA and R75A HDV RNA virions. Although this observation suggests that the R75A mutation does not affect, or at least does not affect to a significant extent, the formation of the HDV RNP, we cannot formally eliminate the possibility that R75A S-HDAg could be at least partially impaired for its binding capacity to viral RNA. However, we believe that it is unlikely since the main RNA-binding site is located the N-Terminal part of the protein (AA 2-27) well upstream of both the R75A mutation and upstream of the NLS. The new results obtained in HDV infection models with the anti-BAZ2B shRNAs (new Figure 2b) and with the BAZ2B BRD inhibitor GSK-2801 (new Fig 2c and 2e) underline the role of S-HDAg interaction with the BAZ2B BRD for HDV replication and support the interpretation that the loss of R75A S-HDAg interaction with the BAZ2B BRD is the prominent mechanism for the reduced replication of the R75A HDV mutant virus.

Question 7. Fig. 4a: bar charts, y-axis: the "1" in "100" of your scale is missing

Answer 7. The new Fig 4a in the revised manuscript is now correct.

Question 8. Page 14, line 11: delete "s" from "interacts"

Answer 8. The paragraph discussing some of the ITC results where the words 'might freely interacts' appeared has been withdrawn in the revised manuscript (see also the answer to *Question 8* from *Reviewer 2*).

Reviewers' comments:

Reviewer #1 (Remarks to the Author):

Well revised, No further comments

Reviewer #2 (Remarks to the Author):

The revised manuscript is considerably better and more convincing. However, the conclusions are still somewhat overstated.

Minor revisions:

- Sup 1d – This image is too dark to see nuclear accumulation and also very zoomed out. Staining of a cytoplasmic to compare would be helpful. A merged image (as in 3b) would also help.

- Sup 1e – The negative control with the D2M plasmid in HepaRGs without dox to show that this plasmid is deficient in this system is missing. This was shown in the original submission and should not have been removed.

- Fig 2a-b – Does shRNA KD of BAZ2B result in any cellular toxicity that would explain the decrease in viral replication? If the cells are dying, the virus would be expected to do worse. Authors showed cell viability for the GSK compounds but not the KD.

- Line 203 – why is the ST-S-HDAg only partially purified? My understanding is that the beads with the streptavidin tagged S-HDAg were then directly incubated with the recombinant proteins, but this is not entirely clear.

- Fig 3a – in the consensus motif, the XX residues vary considerably from those of the H3 tail or SNF2 sequences, i.e. a proline is structurally distinct from an alanine, and a hydrophobic residue from a charged residue. Although these are not the focus, they may represent other possible regulations that should be addressed in the text.

- Fig 3e and 4 – although I agree with the authors that their results support that the R75A mutation interrupts BAZ2B binding, it is not clear that this is due to direct binding through the purported SLiM motif. I appreciate that the authors have removed the results from the calorimetric assays to not confuse readers, however, the new figure 3 still does not support conclusive direct binding as the

change from a charged arginine to an alanine could have other effects on the protein structure. Could this be destabilizing the entire protein and therefore non-specific? Thus, the title of figure 4 is overstated. The data support that the R75A mutation causes a (mild) defect in HDV fitness, but it is not clear that this is solely due to disrupting BAZ2B binding.

- The conclusions are also somewhat overstated in that the authors state 'we found that the lack of interaction between the S-HDAg R75A and BRFs is associated with a reduced level of HDV RNA replication' when they did not show a decrease in association of the mutated S-HDAg with the BRFs during infection. They showed the interactions were disrupted in ectopic expression of the viral protein and with tagged BAZ2B, but not during HDV infection itself. This is a minor jump forward, but should not be overstated in the discussion. The same regarding Pol II recruitment which has only been shown indirectly. The authors should state that their model is proposed based on their findings.

Reviewer #3 (Remarks to the Author):

The authors have adequately addressed the concerns I had raised during the first review. The new virological data supports their hypothesis and improves the manuscript significantly.

Point to point response to reviewers' comments

Reviewer #1.

Well revised, No further comments

We thank again the reviewer for the favorable evaluation of our work.

Reviewer #2.

General comment

The revised manuscript is considerably better and more convincing. However, the conclusions are still somewhat overstated.

We have been happy to learn that Reviewer 2 found our revised MS “*considerably better and more convincing*”. We have now revised the MS according to the criticisms from the Reviewer and we performed additional experiments to assess:

- a) PHH survival following BAZ2B shRNAs transduction in non-infected and HDV-infected cells (new Figure 2c);
- b) WT S-HDAg and R75A S-HDAg protein stability (new Figure 3e);
- c) BRG1 components and Pol II recruitment onto the HDV RNP in PHHs infected with wt HDV or the recombinant R75A HDV mutant (new figure 4c);
- d) the purity of ST-S-HDAg protein preparations by silver staining (Supplementary Fig. 4a);
- e) the specific interaction between ST-S-HDAg and Ni-NTA-bound His-Tag BAZ2B BRD in reciprocal pull-down assays (Supplementary Fig. 4c).

Minor points

Question 1. *Sup 1d. This image is too dark to see nuclear accumulation and also very zoomed out. Staining of a cytoplasmic to compare would be helpful. A merged image (as in 3b) would also help.*

Answer 1. Supplementary Fig. 1d has been improved and both insets with *zoom in* images and merge images have been included.

Question 2. *Sup 1e. The negative control with the D2M plasmid in HepaRGs without dox to show that this plasmid is deficient in this system is missing. This was shown in the original submission and should not have been removed.*

Answer 2. Supplementary Fig. 1e has been reworked into 2 independent Figures (Supplementary Figure 1e and Supplementary Figure 1f) to show that the pSVL-D2M is defective. In Fig. S1e we confirm that pSVL-D2M is defective in HepaRG cells (lane 3) and we show that ST-S-HDAg *trans*-complements pSVL-D2M upon doxycycline induction in the HepaRG stable cell line (lane 2) as well as upon co-transfection in HepaRG cells (compare ST-S-HDAg + pSVL-D2M in lane 4 vs pSVL-D2M alone in lane 3). Similar results were obtained in Huh7 cells co-transfected with pSVL-D2M and pEXPR_ST_S-HDAg (Fig. S1f; compare ST-S-HDAg + pSVL-D2M in lane 2 vs pSVL-D2M alone in lane 1).

Question 3. *Fig 2a-b. Does shRNA KD of BAZ2B result in any cellular toxicity that would explain the decrease in viral replication? If the cells are dying, the virus would be expected to do worse. Authors showed cell viability for the GSK compounds but not the KD.*

Answer 3. A new panel has been added (Figure 2c) to show PHHs survival following BAZ2B and scrambled shRNAs transduction in non-infected and HDV-infected cells.

Question 4. *Line 203 – why is the ST-S-HDAg only partially purified? My understanding is that the beads with the streptavidin tagged S-HDAg were then directly incubated with the recombinant proteins, but this is not entirely clear.*

Answer 4. We apologize if our wording (“*partially purified*”) was somehow misleading. We have revised the Results section (page 7) with the description of the *in vitro* pull-down experiments performed to evaluate the direct interaction between S-HDAg and BAZ2B BRD and we have performed an additional, reciprocal, pull down experiment (see below, the description of the new Supplementary Figure 4c in the Results section and the pull down assays description in the material and methods at page 19). The Strep-tagged ST-S-HDAg was expressed in Huh7 cells and purified by

affinity chromatography using StrepTactin® magnetic beads in the presence of Sodium butyrate to preserve S-HDAg acetylation, that is required for Pol II-mediated HDV genomic RNA synthesis and HDAG mRNA transcription. The purity of the StrepTactin®-bound ST-S-HDAg protein was verified by silver staining (now shown in the new Supplementary Figure 4a). Supplementary Figure 4b shows that ST-S-HDAg and not to the untagged S-HDAg protein or to histone H3 binds specifically to StrepTactin® beads (Supplementary Figure 4b, ex Fig. S4). The *in vitro* pull down was performed by incubating purified ST-S-HDAg with bacterially expressed recombinant His6-BAZ2B-BRD or His6-GFP as a negative control. The eluates were analyzed by immunoblot showing the specific binding of His6-BAZ2B BRD to ST-S-HDAg (Figure 2g, compare lanes 2 and 4) but not to His6-GFP (Figure 2g, compare lanes 1 and 3). In the new Supplementary Figure 4c we show that ST-S-HDAg purified from Huh7 cells using StrepTactin® magnetic beads and subsequently eluted by biotin was specifically retained on a BAZ2B BRD-linked Ni-NTA agarose resin.

Question 5. - Fig 3a – in the consensus motif, the XX residues vary considerably from those of the H3 tail or SNF2 sequences, i.e. a proline is structurally distinct from an alanine, and a hydrophobic residue from a charged residue. Although these are not the focus, they may represent other possible regulations that should be addressed in the text.

Answer 5. In order to answer to the concern raised by Reviewer 2 we have integrated in the Results section (page 8) a more detailed description of the SLiMs in H3 and H4 tails (K14acAPR in H3 and K16acRHR in H4) that interact with the BAZ2B BRD and added in Figure 3a the sequence of the H4 SLiM (see also below). The presence of an acetylated lysine in position 1 of the KacXXR SLiM as well as an arginine at SLiM position 4 (R17 in H3 and R19 in H4) are strict requirements. The second residue of the SLiM motif tolerates wide amino-acid changes whereas there is a strong preference for hydrophobic or aromatic amino acids in the third position of the SLiM. The analysis of 273 HDV isolates (see Figure 3a) shows the presence in position 2 of an arginine (similar to H4) or a lysine in 73.3% and 26.3% of the isolates, respectively. In position 3, proline (hydrophobic) was present in 13.3% of the isolates and substituted by an alanine (hydrophobic) in 81.2% of the isolates. Thus, although we cannot but agree with Reviewer 2 that “a proline is structurally distinct from an alanine” (position 3) and that “a hydrophobic residue .. (such as a proline) is different .. from a charged residue .. (such as a lysine)”, the KacR/kA/pR motif in S-HDAg meets all the predicted criteria to be considered a putative BRD SLiM (i.e. acetylated lysine in position 1, no strict requirement in position 2, hydrophobic or aromatic amino acid in position 3, arginine in position 4) and to mimic the H3 and H4 KacXXR motifs.

It is worthy to underline that the *in vitro* pull down experiments shown in Figure 2g and in Figure S4c together with co-immunoprecipitation experiments shown in the Figure 3f and the new RIP experiments performed in PHHs infected with the R75A virus (Figure 4c), taken together, support the notion of S-HDAg K72acXXR75 sequence acting as a SLiM for the interaction with BAZ2B BRD and the recruitment of BRF complexes on the HDV RNP.

Question 6. - Fig 3e and 4 – although I agree with the authors that their results support that the R75A mutation interrupts BAZ2B binding, it is not clear that this is due to direct binding through the purported SLiM motif. I appreciate that the authors have removed the results from the calorimetric assays to not confuse readers, however, the new figure 3 still does not support conclusive direct binding as the change from a charged arginine to an alanine could have other effects on the protein structure. Could this be destabilizing the entire protein and therefore non-specific? Thus, the title of figure 4 is overstated. The data support that the R75A mutation causes a (mild) defect in HDV fitness, but it is not clear that this is solely due to disrupting BAZ2B binding.

Answer 6. As stated in the answer to Question 5, we think that the *in vitro* pull down experiments (Figures 2g and S4c) together with the co-immunoprecipitation experiments (Figure 3f) and the RIP experiments performed in PHHs infected with the R75A virus do support the notion of S-HDAg K72acXXR75 sequence acting as a SLiM for the interaction with BAZ2B BRD and the recruitment of BRF complexes on the HDV RNP. In order to answer to the concern raised by the Reviewer on the possible impact of the R75A mutation on S-HDAg protein stability, we have performed a Cycloheximide chase experiment to compare the half-life of wt HDAG and the R75A S-HDAg (see new Figure 3e). Thus, according to the data shown in 3b, 3c, 3d and 3e, the R75A mutation does not affect S-HDAg nuclear localization, acetylation and protein stability. However, we agree with Reviewer 2 that we cannot exclude that the R to A substitution might affect, besides disrupting the binding to BAZ2B and the recruitment of BRF remodelers onto the HDV RNP, other functions of the S-HDAg protein that may also impact on HDV replication. We have modified the text both in the appropriate Result section (page 9) and in the Discussion (page 10) to introduce a note of caution and acknowledge this limitation.

Question 7. - The conclusions are also somewhat overstated in that the authors state ‘we found that the lack of

interaction between the S-HDAg R75A and BRFs is associated with a reduced level of HDV RNA replication' when they did not show a decrease in association of the mutated S-HDAg with the BRFs during infection. They showed the interactions were disrupted in ectopic expression of the viral protein and with tagged BAZ2B, but not during HDV infection itself. This is a minor jump forward, but should not be overstated in the discussion. The same regarding Pol II recruitment which has only been shown indirectly. The authors should state that their model is proposed based on their findings.

Answer 7. The Reviewer raises here an important point. Indeed, in the revised manuscript we showed that the R75A mutation interrupts BAZ2B binding to S-HDAg and that an R75A HDV displays a 1.5 log to 2 log reduction in HDV RNA levels as compared to PHH infected with the wt HDV, BUT we did not show a direct evidence of “a decrease in association of the mutated S-HDAg with the BRFs during infection”. We have performed a new series of RNA immunoprecipitation experiments aimed to compare the recruitment of BRFs components and Pol II on the viral RNP in PHHs infected with a wild type HDV or the R75A mutant virus. The results (new Figure 4c) clearly show that the recruitment of both Pol II and BRF proteins onto the viral RNP is severely impaired, as compared to wt HDV in the context of viral infection in PHHs. We have revised the MS to include and discuss these new results. As already stated in the Answer to Question 6, we have also modified the text both in the appropriate Result section (page 9) and in the Discussion (page 10) to introduce a note of caution and acknowledge Reviewer 2 concerns. Finally, we clearly stated in the Discussion and in the Legend to Figure 5 that our model is proposed based on the findings reported in this manuscript.

Reviewer #3.

The authors have adequately addressed the concerns I had raised during the first review. The new virological data supports their hypothesis and improves the manuscript significantly.

We are very happy to have answered convincingly to Reviewer 3 concerns and that he found our MS improved significantly.

REVIEWERS' COMMENTS:

Reviewer #2 (Remarks to the Author):

The authors have adequately addressed my concerns from the last revision. The manuscript is much clearer and well executed.